# Early selection of task-relevant features through population gating

Joao Barbosa [1] ✉, Rémi Proville[2], Chris C. Rodgers [3], Michael R. DeWeese[4], Srdjan Ostojic [1] & Yves Boubenec [5]

Brains can gracefully weed out irrelevant stimuli to guide behavior. This feat is believed to rely on a progressive selection of task-relevant stimuli across the cortical hierarchy, but the specific across-area interactions enabling stimulus selection are still unclear. Here, we propose that population gating, occurring within primary auditory cortex (A1) but controlled by top-down inputs from prelimbic region of medial prefrontal cortex (mPFC), can support across-area stimulus selection. Examining single-unit activity recorded while rats performed an auditory context-dependent task, we found that A1 encoded relevant and irrelevant stimuli along a common dimension of its neural space. Yet, the relevant stimulus encoding was enhanced along an extra dimension. In turn, mPFC encoded only the stimulus relevant to the ongoing context. To identify candidate mechanisms for stimulus selection within A1, we reverse-engineered low-rank RNNs trained on a similar task. Our analyses predicted that two context-modulated neural populations gated their preferred stimulus in opposite contexts, which we confirmed in further analyses of A1. Finally, we show in a two-region RNN how population gating within A1 could be controlled by top-down inputs from PFC, enabling flexible across-area communication despite fixed inter-areal connectivity.

The informational value of different stimuli can change dramatically depending on the context, but animals can adapt with impressive flexibility to virtually any contingency change. A classical example of this feat is the so-called "cocktail party effect", which refers to our ability to focus on a specific, currently relevant conversation while ignoring all the others. Understanding how stable neural circuits implement this kind of flexible, context-dependent behavior has proven challenging. While there is a growing consensus that it emerges from the interaction between different regions along the brain hierarchy[1–4], the specific interactions are unclear.

One possibility is that regions early in the hierarchy merely represent the incoming stimuli and propagate their representations downstream, where context-dependent rules are applied to effectively guide

behavior[5–8]. In line with this view, pioneering work combining artificial neural networks and neurophysiological recordings from monkeys performing a canonical context-dependent task[9], shows that both relevant and irrelevant stimuli are encoded as late as the frontal cortex, suggesting that the selection of relevant stimuli indeed may occur late in the cortical hierarchy. Empirical evidence demonstrates however that primary sensory areas are modulated by behavioral context[4,10–13], potentially through feedback interactions with downstream areas that could control the selection of the relevant stimulus upstream[14,15]. This evidence supports early models of parallel distributed processing[16], that proposed that task-relevant stimuli encoding could be enhanced by top-down inputs to sensory neurons. The prefrontal cortex[17] is deemed essential in providing these inputs, which push task-irrelevant units to

[1]Laboratoire de Neurosciences Cognitives et Computationnelles, INSERM U960, Ecole Normale Superieure - PSL Research University, 75005 Paris, France. [2]Tailored Data Solutions, 192 Cours Gambetta, 84300 Cavaillon, France. [3]Department of Neurosurgery, Emory University, Atlanta, GA 30033, USA. [4]Department of Physics, Helen Wills Neuroscience Institute, and Redwood Center for Theoretical Neuroscience, University of California, Berkeley, CA, USA. [5]Laboratoire des Systèmes Perceptifs, Département d'Études Cognitives, École Normale Supérieure PSL Research University, CNRS, Paris, France. ✉e-mail: palerma@gmail.com

the low-gain region of their dynamic range, thereby effectively reducing their sensitivity. While an attractive possibility, the specific mechanisms through which different cortical areas cooperate to select the relevant stimuli earlier in the cortex are unclear.

Here, we examine the population dynamics in the rat primary auditory cortex (A1) and the prelimbic region of medial prefrontal cortex (PFC), and propose a mechanism through which interactions between these two areas flexibly select relevant stimuli within A1 in a context-dependent task[13]. We found that both relevant and irrelevant stimuli were encoded within a sensory subspace of A1, in line with other studies of humans and other animals performing context-dependent tasks[2,4,13]. However, we found that the relevant stimuli were furthermore projected along an additional dimension, which we named 'selection axis'. On the other hand, PFC encoded only the decision, fully determined by the selected stimuli. Both areas encoded context robustly throughout the trial. To investigate how this contextual information could drive stimulus selection in A1, we trained recurrent neural networks (RNN) on a similar task. Using the same analyses, we found that the geometry of the relevant and irrelevant stimuli representations resembled those of the rat's A1. Reverse-engineering the mechanisms employed by these networks[18–20] predicted that context-modulated populations selectively gate the relevant stimuli in a context-dependent fashion, with different populations selecting specific stimuli in their preferred context. Further analyses of neural recordings revealed a similar population structure in A1, validating the model prediction and suggesting it could subserve the flexible communication of the selected stimulus with mPFC.

A possible interpretation of our within-area modeling and data analyses is that context-dependent gain modulation occurring within A1 could be controlled by top-down inputs from PFC[16,17]. A recent hypothesis posits that different regions communicate through low-dimensional subspaces[21–23], but how the information being communicated could alternate flexibly to solve a context-dependent task is unclear. Our final contribution is to show through network modeling that within-area gain modulation[19], controlled by across-area inputs, could sub-serve such flexible communication along low-dimensional subspaces. Specifically, we demonstrate that a previously proposed class of RNNs constrained to have within-area low-dimensional dynamics[18–20] can be naturally extended to account for across-area communication subspaces. In a two-region RNN, we show that relevant stimuli information can be transmitted between A1 and PFC in a context-dependent manner, despite fixed inter-area connectivity. Our model is a neural implementation of the communication subspace hypothesis[22,23] that solves a cognitive task and suggests a specific mechanism through which areas could interact flexibly along fixed connectivity subspaces.

## Results

### Context-dependent stimulus representations in A1

To investigate how relevant stimuli are selected to guide flexible behavior, we analyzed neural activity previously collected[13] while rats performed a context-dependent, go/no-go auditory task (Methods). The animals were presented with an auditory stimulus (250 ms) consisting of a pitch warble from both speakers mixed with a broad-band noise lateralized to just one speaker (Fig. 1a, left). Contexts were alternated in blocks and indicated the relevant stimulus feature, i.e. pitch level (high or low) or noise location (left or right). The relevant feature (e.g. left/right in the location task; Fig. 1a) indicated to the animal which port it had to lick to obtain a reward in each context (e.g. go-left/no-go, in the location task; Fig. 1a). Before contextual block changes, the rats performed 20 "cue trials", in which the rat heard only relevant sounds without the irrelevant feature. Single-unit spike trains were collected either from the primary auditory cortex (A1) or medial prefrontal cortex (PFC) while the animals performed the task (Methods). In this study we focused on pseudo-trials (Methods) but results

were qualitatively similar when analyzing simultaneously recorded neurons (Fig. S6). For a detailed description of the training procedure as well as behavioral analyses after training, we refer the reader to the original publication[13].

Previous decoding analyses of this dataset[13] showed that A1 represents the ongoing context (Fig. S1) and both stimulus features, regardless of their behavioral relevance (Fig. 1a, bottom). The specific encoding format of these features across different contexts was not examined, however. Here, we investigated if a given feature (pitch or localization) was encoded in the same format across contexts[24–27], i.e. independently on whether it was relevant ("relevant context") or irrelevant ("irrelevant context"). In Fig. 1b, we illustrate three possible encoding scenarios in the neural activity state space, where each dimension represents a different neuron: (i) *identical encoding*, where the coding axes for the same feature are parallel between the relevant and irrelevant contexts; (ii) *selection encoding*, where the relevant go stimulus is enhanced by adding activity along a selection axis; and (iii) *independent encoding*, corresponding to orthogonal coding axes for the same feature across relevant and irrelevant contexts. If an auditory feature is encoded in similar formats across contexts (*identical*, Fig. 1b), projecting the activity collected during one context onto the decoding axis determined in the other context leads to similar separability between conditions (*identical* in Fig. 1b, bottom). On the other extreme, if the same feature is encoded in orthogonal formats in the two contexts, across-context projections are not separable (*independent* in Fig. 1b, bottom). In between these two extremes, for selection encoding, the two conditions are equally separable along the decoding axes determined in the irrelevant context (Fig. 1b, sensory axis), but not as much along the decoder determined in the relevant context (*selected* in Fig. 1.b, bottom). Note that the three scenarios detailed here do not provide an exhaustive list of all the possible encoding geometries (see Fig. S1c for two other scenarios). Importantly however, each encoding geometry is fully characterized by the angle between relevant and irrelevant axis and their across-context decoding performance (Fig. 1b).

To distinguish these possibilities, we trained stimuli-decoders on trials collected during the irrelevant context and tested their performance on trials during either context. We found that decoders trained on irrelevant trials performed well in both relevant and irrelevant contexts (Fig. 1c, left panel), evidence against an independent code and instead suggesting a sensory axis (Fig. 1b) that is shared across contexts. In contrast, the decoding accuracy of irrelevant trials was substantially reduced when tested with relevant decoders (Fig. 1c, right panel), discarding an identical code and suggesting a selection axis (Fig. 1b) along which a specific condition was enhanced in the relevant context. We quantified the angle between relevant and irrelevant decoding axes and found that they were aligned, but not parallel, as expected in a selection code (Fig. 1d, e, insets on the left). We therefore estimated the selection axis as the component of the relevant decoding axis that was orthogonal to the sensory axis (Fig. 1b). To visualize this particular encoding geometry, we then projected the trajectories of activity elicited by identical stimuli in the two contexts along the selection and sensory axes (*Across-context decoding* in Methods). We found that stimuli elicited activity mostly along the sensory axis when the stimuli were irrelevant (Fig. 1d, e, gray lines), but also along the selection axis when the same stimuli were presented in the relevant context (Fig. 1d, e, orange lines). To maximize the separation between go and no-go trials, we performed the decoding analyses separately for each context, but a common selection axis across context could be found with similar dynamics (Fig. S1d).

To elucidate how these context-dependent transformations could emerge in A1, we next trained a single-area RNN in a similar task.

### Single-area RNN predicts a non-random population structure

We implemented the context-dependent task of ref. 13 using the NeuroGym toolbox[28]. Our task was similar to those of previous

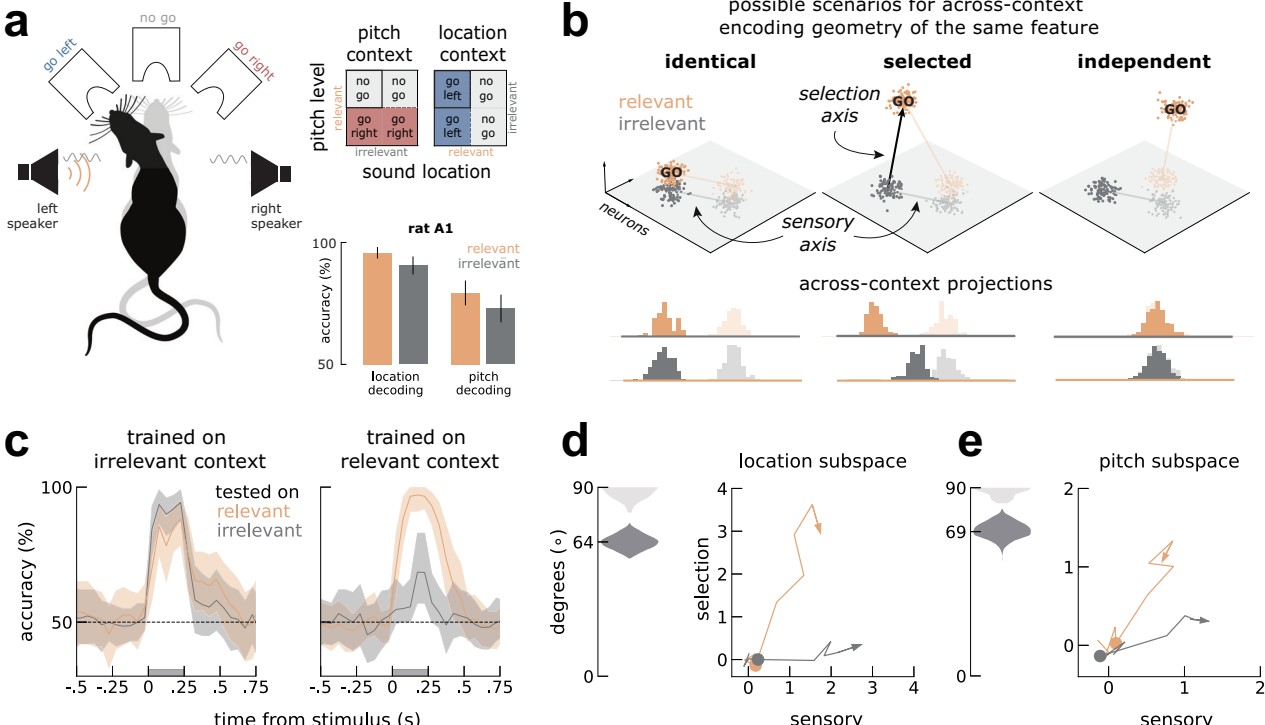

**Fig. 1 | Relevant and irrelevant stimuli are encoded in different subspaces in A1.**
**a** Left, schematics of the auditory discrimination go/no-go task. Rats were presented with an auditory stimulus with two features (pitch and location). Two example trials (black vs gray rat) for the same stimuli (in orange, a noise burst on the left speaker and in gray a high pitch warble on both speakers) in different contexts (location, black; pitch, gray). Depending on the context, the animals had to attend to one of the stimulus features and respond accordingly: go left in location context or no go in the pich context, for this stimulus pair. Right top, context-dependent go/no-go task rules specifying correct behavior for all stimulus pairs. Highlighted (black box) is the stimulus pair illustrated on the left. Bottom, both features are significantly decodable from A1, whether relevant (orange) or irrelevant (gray)[13]. **b** Three possible scenarios for the encoding of the same feature depending on its relevance in each of the two contexts (orange and gray), as characterized by the geometric relationship of the coding axes across contexts. Different transparency levels refer to different conditions (e.g. left vs right location). Left: Identical encoding, where the coding axes are parallel in the two contexts; middle: enhanced encoding, where the go stimulus is enhanced by adding activity along a selection axis; right: independent encoding, corresponding to orthogonal coding axes. Bottom: to distinguish between scenarios, we project the activity of the same stimuli separately when they were relevant (Fig. 2b,

trials in one context (colored histograms) onto the decoding axis (colored line) determined in the other context and inspect the resulting discrimination performance. **c** Across-context decoding (*Across-context decoding* in Methods) of location during pitch context and during location context. Left, irrelevant decoders work well both on irrelevant (gray) and relevant trials (orange). Right, relevant decoders work substantially better in relevant trials than in irrelevant trials. Shaded area marks the stimulus presentation period. See Fig. S5a for similar analyses on pitch trials. **d** On the left, the angles between sensory and relevant axis before orthogonalization, estimated during location blocks are shown in dark gray; for comparison, angles between random vectors (computed by shuffling the weights of each neuron) are shown in light gray. On the right, visualization of the activity elicited by relevant and irrelevant stimuli within the sensory-selection subspace after orthogonalization (Methods). Colored circles mark the stimulus onset. **e** Same as (**d**), but for the pitch context. See also *Slight asymmetry between contexts in* Supplementary Notes. All decoding accuracy computed as the mean accuracy across n=1000 pseudo-populations (Methods). All error-bars are bootstrapped 95% C.I (n=1000 bootstraps). Rat illustration in a) taken from Costa, Gil. (2020). Rat from the top. Zenodo. https://doi.org/10.5281/zenodo.3926343.

studies[9,19], but with a different output space consisting of 3 possible actions, two of them activated in each context. Stimuli (A and B) were delivered transiently (gray bar, Fig. 2c), while the context was delivered throughout the whole trial (Methods). Aiming to replicate the stimulus selection seen in A1, we trained the network to select the relevant go stimulus along a readout vector that was fixed across contexts. To mimic our observations of mixed selectivity in A1 (Fig. S1b), the stimuli and the readout weights on individual neurons were generated randomly[29] and fixed during training (*Trained A1 network* in Methods).

To obtain easily interpretable RNNs, we constrained the recurrent connectivity matrix to be of low rank, allowing us to reverse-engineer the mechanisms employed by the trained networks[18,19,30]. We found that a rank-one network was able to solve the task (Fig. 2a, bottom left; Fig. S9), so that the connectivity matrix was defined by the outer product of two vectors, the output and input-selection vectors (*Low-rank theory* in Methods). After training, we froze the weights and collected the dynamics of all units during all types of trials. As we did with the biological units recorded from A1 (Fig. 1d, e), we projected the activity of the same stimuli separately when they were relevant (Fig. 2b,

orange) or irrelevant (Fig. 2b, gray) onto the output and sensory axes (*Trained A1 network* in Methods). This confirmed that, as in A1, the network represented both stimuli along the same sensory axis, independently on whether they were relevant or irrelevant in the current context, but the relevant go stimulus was enhanced along an additional axis.

We then used recently developed methods to reverse-engineer the mechanism through which the network learned to solve the task. Recent theoretical work has shown that context-dependent tasks such as the one considered here require neurons to be organized in different populations, each characterized by its joint statistics of connectivity parameters[18,19]. A key empirical test of this finding is that networks with shuffled connectivity parameters should still solve the task with an accuracy similar to trained ones, as long as the statistics of connectivity within each population are preserved[19]. Performing this analysis (*Inferring populations* in Methods), we found that our trained networks relied on at least three populations, as resampling the connectivity vectors from less than three populations did not achieve high performance (Fig. 2a, bottom right). Due to differences in how these

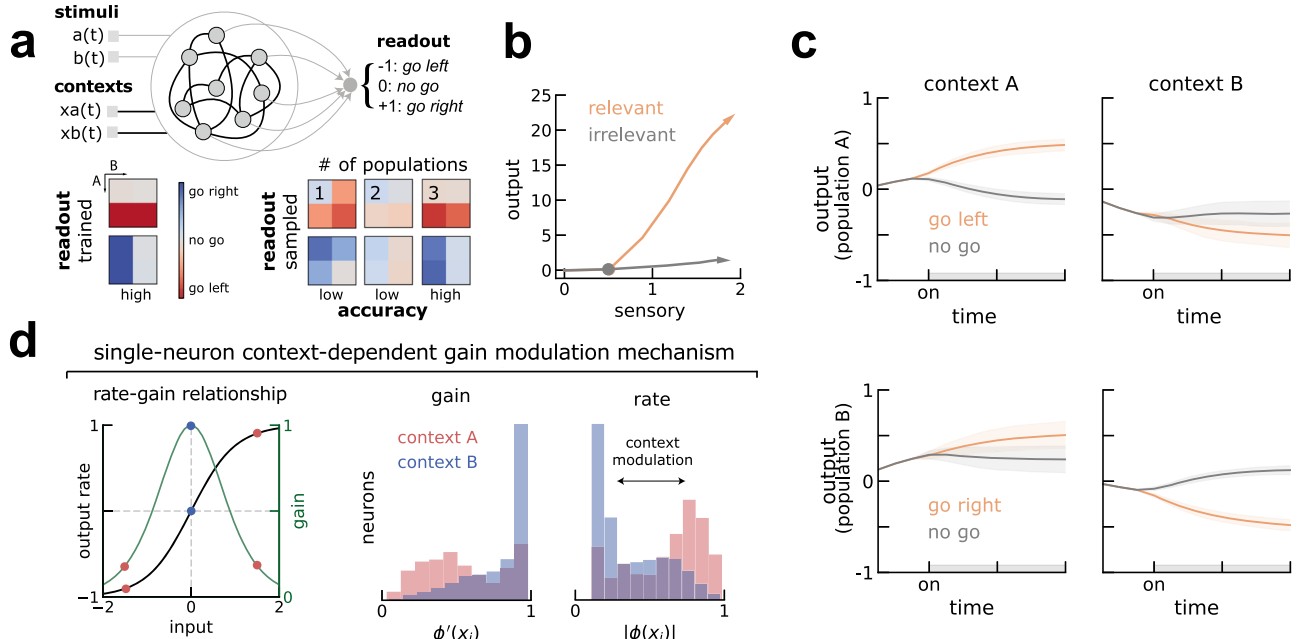

**Fig. 2 | Trained RNN replicates A1 dynamics and predicts population gating supports flexible selection of the relevant stimuli. a** Top, schematics of the RNN. On each trial, the RNN receives 4 inputs (stimuli and contexts) and must output the correct choice (−1, 0 or +1, representing go left, no go or go right) onto a fixed readout axis. Depicted in black are the weights that are trained with back-propagation (i.e. contextual inputs and recurrent weights) and in gray those that remain fixed (i.e. stimuli input and readout). Bottom, average responses of trained (left) and resampled (right) networks separated by conditions and context (compare with schematics in Fig. 1a). Left, trained networks achieve perfect accuracy in both contexts. Right, clustering and shuffling the connectivity (*Inferring populations* in Methods) keeping an increasing number of populations shows at least 3 populations (population A, B and 0) are necessary to solve the task with comparable accuracy to trained networks (left). **b)** Similar to A1, the network represents

both stimuli but enhances the relevant go stimuli along an additional axis. Colored circles mark the stimulus onset. See also Fig. 1d, e. **c** Dynamics of activity of populations A and B projected on the output axis is reduced in opposite contexts, effectively gating the relevant go stimulus into the output axis (**b**). Error-bars are standard deviations from the mean. **d** Left, single neurons in each population have different gain levels (left, $\phi'$) in the two contexts (here shown only for population B). Middle, illustration of the single-neuron gain modulation mechanism. Neurons in population B receive strong contextual inputs (see also Fig. S2) in context A (red) that shift the working point of individual neurons on their input-output function to a low gain regime. Conversely for context B (blue). Thus, gain modulation is also reflected in the single-neuron firing rate before the stimulus (right).

populations select distinct stimuli that we describe below, we labeled these 3 populations post hoc as A, B and 0.

We explored the contribution of each population to the overall dynamics leading to stimulus-selection along the output axis. With this aim, we examined separately the dynamics of the 3 populations individually by projecting their activity on the output axis in the two contexts. We observed that two out of three populations showed different stimulus-specific dynamics in the two contexts (Figs. 2c and S2). While population A selected the go stimulus A along the output axis during context A, it did not select this stimulus during context B (Fig. 2c, top), and vice versa for population and stimulus B (Fig. 2c, bottom). Note that neurons within each population were selective to both stimuli along the sensory axis, but collectively selected the go stimulus along an additional axis. We quantified this context-dependent dynamics by computing the context-dependent activation of each population (output gating, Methods) and compared it against randomly chosen populations. We found that populations A and B showed substantially more context-dependent activity along the readout axis than population 0, whose dynamics were not different from a randomly selected population of neurons (Fig. S2d). As found in a recent study[19], context-dependent modulation at the population level relies on selective gain modulation at the single-neuron level. This gain modulation is determined by the working point of single neurons, with neurons with higher firing rate exhibiting lower gain (Fig. 2d, left). For neurons in population B, we calculated their gain as the slope of the transfer function before the stimulus presentation (Fig. 2d, middle Methods) and the corresponding firing rate (Fig. 2d, right) in each context. While neurons from population B were operating at higher

levels of gain during context B, their gain was much lower during context A. In turn, population 0 did not show any gain modulation (Fig. S2d).

These analyses point to population gating through gain modulation as a candidate mechanism for solving the context-dependent task. To test whether population gating selects stimuli in the neural data, we sought a procedure to identify the two relevant populations from single-unit recordings. In the network model, the two populations are characterized by their connectivity and gain modulation, but this information is not directly accessible from extracellular recordings. However, since gain modulation arises from contextual inputs that shift the working point of individual neurons on their input-output function (Fig. 2d, middle; see also different context weight strengths to each population in Fig. S2c), we found that gain modulation was reflected in the neuron's firing rate before stimulus onset (Fig. 2d, right). Specifically, we found that the two key populations in the model had decreased pre-stimulus firing rates in their preferred context. Therefore, the network model predicted that the single-neuron pre-stimulus firing rate would allow us to discriminate between neurons that perform the stimulus selection in the two contexts. We next tested this prediction in A1 data.

## Pre-stimulus activity of A1 neurons predicts their population structure

To test the prediction of different context-modulated neuronal populations selecting different stimuli, we grouped all the neurons recorded in A1 ($n = 130$) based on their context modulation during the pre-stimulus period (Fig. 3a, Mann-Whitney U test corrected for

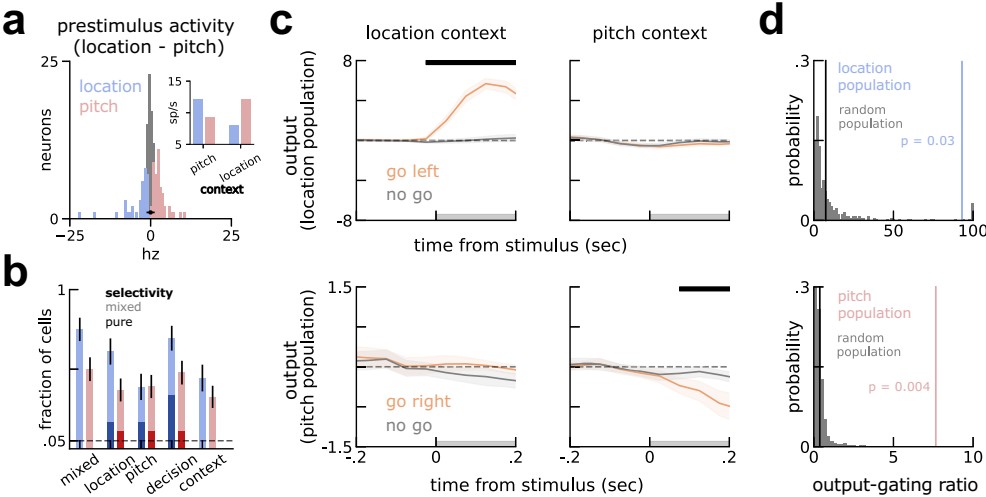

**Fig. 3 | Pre-stimulus context-dependent activity reveals *three distinct populations* in A1, as predicted by the model. a** Neurons are grouped in three populations (location in blue, pitch in red and population 0 in gray), based on their prestimulus firing rate modulation to context. In black, the mean (circle) and [2.5, 97.5] percentiles (bar) of firing rate modulation to context of shuffled trials within neurons. Inset, absolute values of prestimulus activity of location and pitch populations in either context. **b** Neurons in the pitch ($n = 36$) and location ($n = 48$) population (light blue and red bars, respectively) are mixed selective (*Single-neuron selectivity* in Methods), although a small fraction showed pure selectivity to some of the task variables (darker colors). Error-bars are standard errors of the mean. **c** Projection of stimulus responses of go left / right (orange) and no go (gray) onto the output axis

for the location population (top) and pitch population (bottom). Different populations select the relevant go stimuli in different contexts. Left, projections of trials recorded during the location context. Right, equivalently for the pitch context. See Fig. S3 for the projections of individual stimuli, here grouped as go left/right and no go and *Slight asymmetry between contexts* in Supplementary Notes. **d** Permutation test shows output gating in (**c**) is not visible in randomly picked populations (gray, Methods). Top, location-population has an output gating ratio (blue vertical bar) higher than chance ($p = 0.03$). Bottom, pitch-population has gain modulation higher than chance ($p = 0.004$). Population 0 (black vertical bar in both plots) did not show above chance gain modulation ($p > 0.25$). All error-bars are bootstrapped standard errors of the mean[60].

multiple comparisons; see also Fig. S7 for a similar distribution in the models). About a third of the neurons ($n = 48/130$) showed significantly lower spontaneous activity in the location context (henceforth location population) while another third ($n = 36/130$) showed decreased spontaneous activity in the pitch context (pitch population). The remaining ($n = 46/130$) neurons were not significantly modulated by context during the pre-stimulus (population 0). Moreover, grouping neurons by their modulation to context did not separate neurons with opposite stimulus selectivity. Instead, neurons in either population had non-random mixed selectivity to both stimuli (Fig. 3b).

As in the model, we inspected separately the activity of each of the two context-modulated populations projected on the output axis. We estimated the output axis by decoding the two possible outputs (go left vs go right; Fig. 1a and Methods) from each population and projected its activity along this axis, grouping trials by their context and correct output. As predicted by the model (Fig. 2), we found that the population of neurons showing low spontaneous activity in a specific context gated the relevant go stimulus and ignored the irrelevant stimuli (Fig. 3c, top left, and bottom right). Conversely, in the opposite context, the output projection of the same population was essentially identical for all conditions (Fig. 3c, top right, and bottom left; see also Fig. S3). This link between context modulation and stimulus-related dynamics is not trivial, as we grouped neurons during different time points (before and during stimulus) and compared them along different variables (context and stimulus). As done with the simulations, we quantified the level of context-dependent population dynamics (output gating, Methods) and compared it with randomly selected populations. We found that the output gating of both populations ($p = 0.03$ and $p = 0.004$, location and pitch populations, respectively), but not population 0 ($p > 0.25$) was significantly higher than in randomly selected populations (Fig. 3d, Methods). Moreover, the location population output-gating strength was significantly higher than population 0 ($p < 0.0025$) and similarly for the pitch population, albeit not significant ($p < 0.075$).

In sum, we found that neurons grouped by their pre-stimulus context-modulation collectively select different stimuli, as was predicted

by reverse-engineered RNNs. Specifically, individual populations in A1 output the go stimuli in their preferred context but do not in the opposite context.

### mPFC encodes only the relevant stimulus along a selection axis

After characterizing a potential mechanism for the stimulus gating observed in A1, we investigated the existence of context-dependent neural dynamics within mPFC. Previous work has shown causal involvement of mPFC in action-selection during flexible behavior ([13,31]), so we expected to see strong encoding of the stimulus relevant for the decision. As similarly done in A1, we tried decoding both relevant and irrelevant stimuli (Methods). mPFC indeed encoded the relevant stimulus (Fig. 4a, left orange), but it did not encode the irrelevant stimulus (Fig. 4a, gray). We also visualized mPFC context-dependent dynamics along a sensory and selection axis, estimated similarly to A1 (Fig. 1; *Population decoding* in Methods). In contrast to A1, mPFC dynamics evolved exclusively along the selection axis and only very weakly along the sensory axis during the irrelevant context (Fig. 4b). Notably, decorrelating the animal's choice and the relevant stimuli with error trials, we found that both PFC and A1 encoded the relevant stimulus above and beyond choice (Fig. S5c). Furthermore, separating neurons according to their pre-stimulus activity when their activity was strongly selective to context (Fig. S1) did not reveal robust population gating ($p > 0.1$, $p = 0.22$, $p = 0.083$ for population 0, 1 and 2, respectively; output-gating permutation test; Methods. See also Fig. S8). In sum, PFC neurons encode context (Fig. S1) and decision along orthogonal directions (Fig. 4). In contrast to A1, they do not encode the irrelevant stimulus and their activity seems to be well described by one population. Next, we incorporate these observations in a multi-area network.

### Multi-area RNN with across-area population gating replicates A1 and PFC dynamics

Our within-area modeling and data analyses suggest that gain modulation occurring within A1 could be controlled by top-down

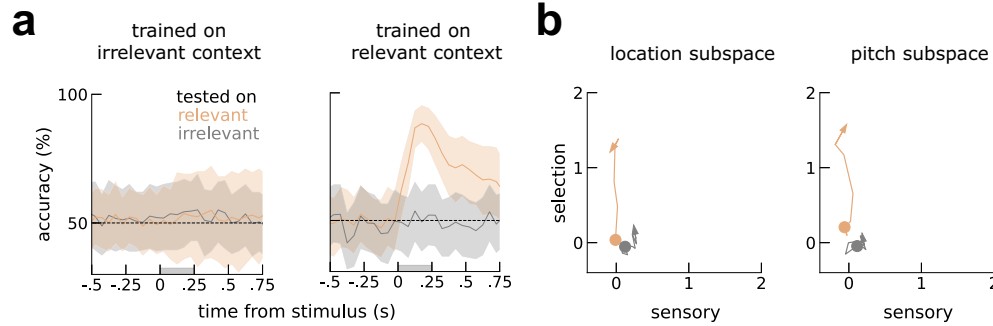

**Fig. 4 | PFC encodes only the selected stimuli along the selection axis. a** Across-context decoding (*Across-context decoding* in Methods) of location during pitch context and during location context. Left, irrelevant decoders fail both on irrelevant (gray) and relevant trials (orange). Right, relevant decoders work well in relevant trials but not irrelevant trials. Shaded area marks the stimulus presentation period. Decoding accuracy computed as the mean accuracy across n=1000 pseudo-populations (Methods). Error-bars are bootstrap 95% C.I. See also Fig. 1c. **b** visualization of the activity elicited by relevant and irrelevant stimuli within the sensory-selection subspace. Colored circles mark the stimulus onset. See also Fig. 1d, e.

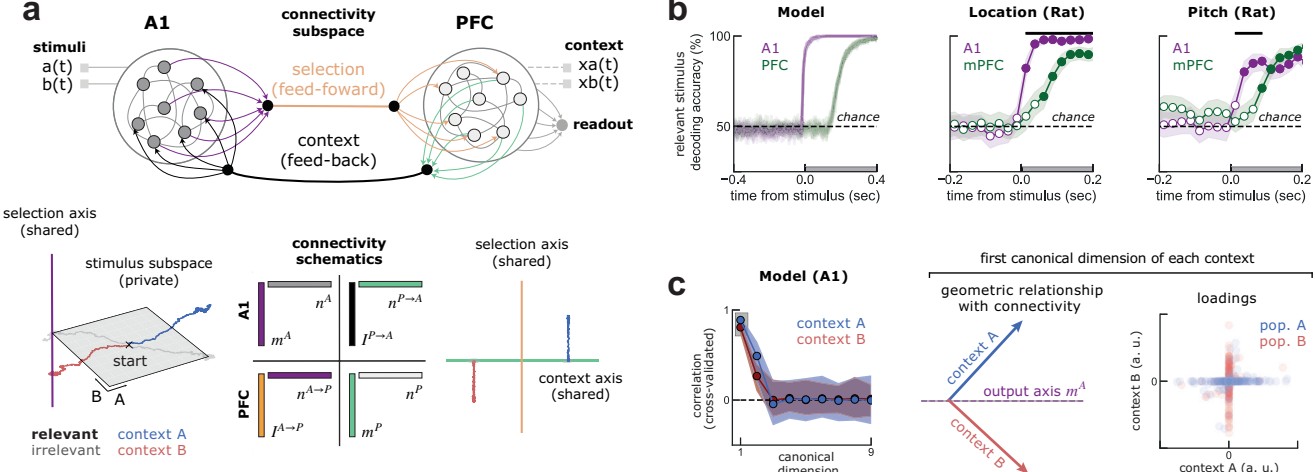

**Fig. 5 | Engineered multi-area model replicates A1 and PFC dynamics and produces predictions for across-area interactions. a** The interaction between A1 and PFC was set to occur through low-rank connectivity in opposite directions (feedforward in orange and feedback in gray). In contrast to the trained network, context is delivered transiently to PFC (dashed), stored in persistent activity and fed back to A1 (context, gray). In turn, stimuli are delivered to A1 and are not communicated to PFC, thus remaining in a "private subspace" of A1 (bottom left). The relevant stimulus, which is selected within A1 by integrating the stimuli and context, is communicated to PFC along the selection axis (selection, orange). Solid black circles represent the two variables that are shared across areas. On the bottom, connectivity schematics illustrates the geometric relationship between within- and across-area connectivity vectors, with similar colors illustrating strong overlap. **b** Relevant stimuli are encoded earlier in A1 than in PFC, as seen in the data for location (middle) and pitch (left). Full circles mark 25 ms bin with decoding significantly above chance ($p < 0.05$, one-sided bootstrap test; not corrected for

multiple comparisons). Black bar marks time bins where A1 decoding is significantly above PFC decoding. Error-bars are SEM. **c** Estimation of the communication subspaces using canonical correlation analyses separately for each context (*Communication subspace estimation* in Methods). On the left, cross-validated correlation along different canonical dimensions is significant for two dimensions (bootstrapped mean and error bars are 95% C.I.). Highlighted in gray is the first canonical dimension of each context, used in the remaining panels. In the middle, we show that the first canonical dimensions of context A and B (left) are orthogonal, i.e. that A1 and PFC communicate through orthogonal subspaces in opposite contexts (red and blue, Methods), despite fixed connectivity. However, we show that these orthogonal subspaces are aligned along the output axis ($m^A$, purple). On the right, we show that these orthogonal subspaces are supported by different populations. Neurons of population A have mostly null coefficients on the communication subspace during context B; conversely for neurons in population B.

contextual inputs. To formally show how across-area interactions can lead to the context-dependent population activity observed during the task, we modeled the within-area dynamics observed in the rat's A1 and PFC in a single model that performed the task (Figs. 5a and S4). For the across-area interactions, our model instantiated the recent hypothesis that different regions communicate through low-dimensional subspaces[21–23]. Under this view, some information within an area is transmitted to a second area through a so-called communication subspace, while the rest remains in a private subspace. While an interesting empirical observation, a concrete network model performing a cognitive task is lacking[21,22]. To directly implement this hypothesis, we engineered a two-region network model by starting

from two low-rank networks representing A1 and PFC, and then connected them by adding low-rank interactions between areas (*A1-PFC network* in Methods). Note that our approach here is in contrast to the single area RNN case (Fig. 2), which are trained. This is because we were interested in exploring the specific hypothesis of flexible communication through across-area gain modulation, rather than generating new hypothesis through network training[32].

Specifically, we set the connectivity geometry of A1 similarly to the trained network (Fig. 2), meaning that when biased by contextual inputs it selected the relevant stimuli along the A1 output axis through population gating (Figs. 2, 3). In turn, PFC was set up to store the current context in persistent activity[20] using only one population, as

suggested by the data analyses. Given our empirical observations (Figs. 1 and 3), our hypothesis was that the sensory stimuli information remains private within A1, while the selected stimulus information is communicated downstream (Fig. 5a). To set up this communication channel (feedforward in Fig. 5a), we implemented the connectivity from A1 to PFC as a rank-one matrix ($J_{A\to P} = I^{A\to P} \otimes n^{A\to P}$), setting $n_{A\to P}$ to be aligned with the output axis of A1 ($m_A$). On the other hand, context is stored in working memory within PFC, but communicated to A1 along another dimension, also implemented by a rank-one matrix ($J_{P\to A} = I^{P\to A} \otimes n^{P\to A}$) representing the connections from PFC to A1 (feedback in Fig. 5a). As in the feedforward case, we set $n^{P\to A}$ to be aligned with the output axis of PFC ($m_A$). This configuration is illustrated in 5a, with similar colors showing axis with strong overlap. Altogether, the connectivity between A1 and PFC was defined by two axes within each area, therefore its dimensionality was $n = 2$.

After setting the two-area network to interact through low-rank subspaces, we tested its performance. This was done similarly to the trained network, but now stimuli were presented only to A1 and context only to PFC. At the end of each trial, we read out the final response from PFC. In isolation, A1 represents all stimuli and PFC stores the current context in persistent activity (Fig. S4a), but they do not select the relevant stimulus in a context-dependent fashion (Fig. S4a, b). When we connect the two areas as described above, contextual information is propagated from PFC to A1, targeting specific neural populations to select the relevant stimulus within A1 (Fig. S4a, bottom). The selected stimulus is then propagated downstream to PFC, from which the final response was read out, effectively solving the context dependent task (Fig. S4b). Due to this model architecture, relevant stimuli are encoded earlier in A1 than in PFC (Fig. 5b, left), as seen in the data (Fig. 5b), even when artificially decorrelating decisions and relevant stimuli with error trials (Fig. S5c).

We then developed predictions about the interaction between A1 and PFC that could potentially be tested in datasets with simultaneously recorded areas. First, we validated a previous approach to estimate communication subspaces in experimental data[23]. To this end, we computed the canonical correlation dimensions (cross-validated, Methods) between A1 and PFC, after removing condition-averaged activity and separating trials by context. Using this approach, we estimated the dimensionality of the communication subspace to be two-dimensional in both contexts (Fig. 5b, left). Interestingly, we found that the networks communicate through context-specific subspaces that are orthogonal to each other (Fig. 5c, middle; Fig. S4b), demonstrating that the communication between A1 and PFC alternates flexibly along different channels in opposite contexts, despite their fixed across-area connectivity. Our model therefore shows how the switch between different communication channels can be controlled by top-down inputs from PFC to A1. Specifically, top-down inputs select which population participates in the communication subspace (Fig. 5c, right), thereby determining what information is selected in A1 for propagation downstream.

All together, our model replicates the dynamics within A1 and PFC and show how across-area population gating can subserve their flexible communication, despite fixed connectivity.

## Discussion

Previous studies of neural activity during context-dependent behavior have found that both relevant and irrelevant stimuli are encoded across the cortex[4,9,13,33]. Here, we used cross-context decoding to characterize the specific encoding geometry of these stimuli in the rat auditory cortex (A1). We found evidence for the selection of the relevant stimuli along an axis ("selection axis") orthogonal to the axis encoding the stimuli ("sensory axis"). This encoding geometry, which we term selection code, is related to previous work on 'cross-condition

encoding'[26]. As it happens, this encoding geometry has several advantages relative to the alternatives illustrated in Fig. 1. First, it allows for sensory information invariance along the sensory axis, even across potentially very different contexts. In view of this, a pear will look like a pear, regardless of your current appetite. Then, despite encoding similar stimuli along a common axis, it allows for their flexible selection depending on their current relevance, in line with the previous findings in the ferret A1 that show that *go* stimuli are enhanced upon task engagement[34].

By reverse-engineering RNNs trained with backpropagation to employ a selection code, we postulate specific mechanisms that could support this code in A1. We found a non-random population structure in the trained RNNs, with two populations selecting different go stimuli. In our model, context-dependent gating of the relevant stimulus was accomplished through gain modulation of specific populations. The model predicted that this population structure could be inferred from pre-stimulus firing rates in electrophysiological recordings. Indeed, we found evidence for such a structure in A1, but not in PFC where the irrelevant stimulus was not encoded. Note that in contrast to our model, which was perfectly symmetric, our decoding analyses of A1 revealed that pitch-related activity was weaker than location, but this does not change the interpretation of our results (see *Slight asymmetry between contexts* in Supplementary Notes).

Our final contribution is to incorporate our empirical findings within A1 and PFC in a multi-area network that postulates their interactions through low-rank communication subspaces. Previous work modeling communication subspaces have focused on noise correlations in spontaneous activity and feedforward interactions[35,36]; see also[37] for a model of 'output-null' subspaces in the context of motor preparation[38]. In contrast, our multi-region network solves a concrete, context-dependent task by setting the areas to interact in both feedforward and feedback directions. Crucially, PFC acts as a controller of A1, dynamically selecting the appropriate communication subspace for the ongoing context. Our model is an explicit implementation of the Miller and Cohen model of PFC[17], (see ref. 39 for a recent review) and complements a large body of computational work focusing on multi-area interactions (see ref. 40 for a recent review). Our major contribution is to propose a single-neuron mechanism (i.e. gain modulation) for flexible selection of different subspaces through population neural dynamics[3,41,42], despite fixed connectivity. We have focused on feature-selection[13], a specific kind of context-dependent tasks[39], and future work will be necessary to extend our results to other types of context-dependent task, such as fear conditioning extinction[43,44] or task switching[39,43,45].

In our model, gain modulation is accomplished by selectively targeting specific units with different contextual inputs (Fig. S2c,[19]), pushing individual units to the non-linear regime of their input-output function. How gain modulation is accomplished in A1 remains to be fully elucidated[46], but possible ways in which neuronal populations in A1 could have reduced gain after increased activity include synaptic non-linearities, such as depression[46], or (loose) balance between inhibitory and excitatory neurons (see ref. 47 for a recent review). Future work is necessary to determine the specific mechanism. In some studies probing context-dependent behavior, actions are decoupled from stimuli with careful task design, while here we tackle this possibility only indirectly (Fig. S5; see also *Discarding motor execution confounds* in Supplementary notes); notwithstanding, the mechanisms proposed here for stimulus selection through gain-modulation would straightforwardly apply to decision selection within A1.

In our model, there is a division of labor: A1 represents all stimuli, but when biased by the current context it selects the relevant stimulus; in turn, PFC reads out the relevant stimulus and uses it to effectively guide behavior and could in principle infer the current context through trial and error. We leave for future work the question of how the current context is inferred. Crucially, these two areas communicate

the key task variables, context and the selected stimulus, through rank-one subspaces in opposite directions. We have assumed that mPFC interacts directly with A1, but the control of stimulus selection in A1 could be accomplished through a third area, such as the thalamus[48] or the amygdala[44]. Furthermore, it is possible that A1 and PFC communicate other variables in addition to, or instead of, ongoing context and the currently selected stimuli. For instance, it is not necessary for PFC to both act as receiver of the relevant go information and providing the context. Both the interaction through a third area and the communication of different variables could be accounted by specific across-area connectivity profiles.

The computational advantages of our model modularity are unclear when considering a single task, as was the focus of this work. Considering instead a more realistic case, in which rats are trained on more than one task, for instance two contexts-dependent task of different modalities (e.g. auditory and visual), the advantages become apparent: in principle, PFC could be reused in both tasks, storing the current context and biasing currently relevant networks. Similarly, A1 could conceivably route different stimuli to another area, instead of only the relevant stimulus to PFC. In this work, we opted to model the simplest hypothesis that could explain our findings within A1 and PFC. Regardless, this model is to our knowledge the first neural implementation of the communication subspace hypothesis[22] that performs a cognitive task (but see ref. [35,36]). While future theoretical work will be necessary to fully flesh out the implications of the communication subspace hypothesis[22,23], our model reveals several interesting insights. First, it shows that communication subspaces, empirically shown to play a role during passive viewing[22,23], can be exploited during flexible behavior. In the model, PFC controls A1 with contextual inputs, biasing it to select and communicate the relevant stimulus. Using canonical correlations analyses (CCA)[22,23], we show that this communication occurs along orthogonal subspaces that are explored flexibly in different contexts, but within the fixed subspace set by the network connectivity. This analysis is a specific prediction that can be tested in multi-area recordings from animals performing context-dependent behavior. Second, while the subspaces estimated with CCA are aligned with those implemented in the model, we used decoding analyses as baselines to show that this estimation is imperfect, mixing feedforward and feedback communication along the same dimensions (Fig. S4c).

With the advent of large-scale recordings, it is becoming clear that animal behavior implicates multiple areas. In a rare tour-de-force, a recent study recorded simultaneously from six areas along the primate visual pathway, while subjects were engaged in a visual context-dependent task[1,4]. This study shows clearly, perhaps unsurprisingly, that visual sensory information is quickly and more strongly encoded in the visual cortex (V4) compared to associative areas, indicative of the feedforward flow of sensory information. On the other hand, the current context and the monkey's decision are encoded earlier and more prominently in higher-order areas, such as PFC[4], consistent with the feedback flow of contextual information in our model. Both types of information are eventually encoded in all of the recorded areas, suggesting inter-area communication in feedforward and feedback directions. Interestingly, both relevant and irrelevant stimuli were decoded across the brain hierarchy, generalizing a previous finding in the monkey frontal eye field[9,33]. In contrast, we did not find encoding of irrelevant stimuli in mPFC, consistent instead with early selection of the relevant stimuli. This discrepancy might be due first and foremost to differences in animal species, but also to task differences. However, another recent study[3] recording simultaneously from several areas across the monkey brain (V4, FEF, Parietal, and PFC), shows that visual areas (V4) encode strongly both relevant and irrelevant stimuli, but areas downstream such as FEF or PFC give clear preference to the relevant stimulus and are more predictive of the upcoming action - in line with the view taken here. Similarly, a recent MEG study of humans

performing a context-dependent task[49], shows that decoding of irrelevant features from the dorsal premotor cortex, to which the prelimbic part of the rat mPFC is arguably reminiscent[50], is substantially lower than the decoding of relevant features. Similar findings have also been reported in human fMRI[2]. Our model reflects these empirical findings and proposes that different areas, with different computational roles, could alternate their communication through orthogonal low-rank subspaces[22,23], despite fixed connectivity. Together with recent work on multi-area interactions[40,51] our work motivates an exciting new perspective on previous and future multi-region recordings[21].

## Methods

### Animal training and electrophysiology

All procedures were approved by the Animal Care and Use Committee at the University of California, Berkeley. We are reanalyzing a previously collected dataset, so we are describing the experimental procedures here only briefly. For a complete description, we refer the reader to the original publication[13].

**Task.** Six rats were trained to respond to either of two simultaneously presented sounds, in a context-dependent fashion. The rats initiated each trial by holding their nose in the center port of a three-port behavior box. Each stimulus was 250 ms in duration and consisted of two different features, location and pitch. More specifically, the stimulus consisted of a noise burst played either from the left or right speaker (location feature), and a high or low pitched frequency-modulated tone (pitch feature), played from both speakers simultaneously. The task alternated between blocks of localization and pitch discrimination trials. Each localization block began with 20 "cue trials" in which only the localization stimulus played, followed by 60 trials on which both localization and pitch discrimination stimuli played simultaneously. Pitch discrimination blocks similarly began with 20 cue trials of pitch stimuli, followed by 60 trials of both stimuli simultaneously. Localization and pitch discrimination blocks alternated throughout the entire session. The session lasted for approximately one hour, and rats were allowed to do as many trials as they wished during this period of time. On each contextual block, one of the features was the relevant feature and its value determined the correct response, while the other feature was deemed irrelevant. During localization blocks, the reward could be collected on the left port (go left) when the stimulus was presented on the left speaker; no reward could be collected when the stimulus was presented on the right (no go), but animals were penalized with a timeout if they left the center port. Conversely, during pitch block, the reward could be collected on the right port (go right) when the pitch was of low frequency; no reward could be collected for high frequency (no go). Correct responses were rewarded with water, while mistakes were penalized with a 2–6 s timeout. Incorrect responses and 'cue trials' were excluded from all the analyses.

**Single-unit recordings.** After training, tetrodes were implanted into the rats' brains, targeting either A1 or the prelimbic region of mPFC and single-unit spikes trains were collected while the animals performed the task. Here, we only analyzed units with a sort quality defined as 'great' or 'good' in the CSV file (https://github.com/cxrodgers/Rodgers2014) recorded during at least 10 correct trials of each kind. All analyses were performed on raw spike counts computed within windows of 50ms. See the original publication for more details[13].

### Single-cell analyses

**Single-neuron selectivity.** To estimate single-neuron selectivity, we counted all the single neuron spike while the stimulus was presented (250 ms) and regress them against a linear combination of all task variables of interest[9], namely location, pitch, decision, and context. We

considered a task variable to be significantly encoded by a neuron if its regression weights were significantly different than 0, as accessed with the statsmodels python package. Neurons with only one significant weight were considered to have pure selectivity and otherwise mixed selective (Fig. 3).

In the original study[13], we considered the possibility that a difference in encoding between the blocks might arise from a slow firing rate drift over the entire session. To test for this, we fit a linear model that explained each neuron's response as a combination of block number within the session as well as block identity (localization or pitch discrimination). Only a small minority of neurons (3.5%) showed an apparent difference between the blocks that was explained by a significant effect of block number and not block identity.

**Identification of distinct populations by pre-stimulus modulation.**
To test the RNN prediction laid out in Fig. 2, we averaged each neuron's firing rate before stimulus (1 s) onset separately in each context. For each neuron, we then tested for their different firing rates in the two contexts (Mann-Whitney U test), corrected for multiple comparisons (Benjamini/Hochberg). Out of $n = 130/131$ neurons in A1/PFC, some neurons had significantly lower firing rates during the location context ($n = 48/58$ in A1/PFC), others during the pitch context ($n = 36/36$) and were thus labeled as location and pitch population, respectively. Neurons that did not show significant context modulation ($n = 46/37$) were labeled as "population 0".

## Population analyses

**Pseudo-population decoding.** All decoding analyses were performed on 'pseudo-trials', pooling across all animals[52]. We opted to decode from pseudo-trials to maximize decoding accuracy. This is particular important to support our claims of lack of decodability (e.g. irrelevant decoding in mPFC). Crucially, our results do not depend on these methodological choices and are qualitatively similar when decoding from simultaneously recorded populations of 1 to 13 neurons (Fig. S6). Specifically, we build pseudo-simultaneous populations by resampling with repetition 50 pseudo-trials from each condition and neuron. We repeated this process 500 times, leading to 500 folds across which we computed decoding variability. All decoding performances were cross-validated by splitting the training and testing dataset in two halves (50% trials for testing). Importantly, the dataset splitting was performed independently for each fold. We decoded the variable of interest – context, location or pitch – using the scikit-learn package sklearn.linear_model.LogisticRegression. To estimate the output axis in Fig. 3, we computed the distance between the average activity during go and no-go condition[34]. We then projected the activity separated for each condition along the same axis.

**Across-context decoding.** To investigate the stimuli encoding geometry within and across contexts, we performed across-context decoding[26,27]. In this case, we also used pseudo-populations, but training and testing was done with datasets collected during different contexts. For instance, we trained logistic regression decoders to discriminate the location of the stimuli (left vs right) during pitch (location) blocks and then tested these decoders either on pitch blocks or on location blocks (Fig. 1c). When training and testing within the same context, we set aside 50% of trials for cross-validation. This was not necessary when this process was done across contexts, but we also subsampled 50% of the trials within each fold to avoid unfair comparisons. We repeated this process for all time points (Fig. S5) and found that selection and sensory axes were stable during stimulus presentation. For all decoding analyses we therefore used the average weights during this period.

For the visualization of activity along a decoding axis, we removed the non-linearity of logistic regression. Specifically, we collected the weights trained with logistic regression and projected

the activity elicited by go and no-go stimuli on these weights. We then plotted the distance between these two conditions (without applying the logistic non-linearity). Importantly, before projecting on these weights, we orthogonalized the sensory and selection axis using QR decomposition[9].

**Output-gating permutation test.** To quantify the degree of output gating seen in A1 (Fig. 3), we calculated the following ratio during stimulus:

$$\frac{|GO_{ctx=pop} - NoGO_{ctx=pop}|}{|GO_{ctx\neq pop} - NoGO_{ctx\neq pop}|} \tag{1}$$

With GO (NoGO) corresponding to the average activity elicited along the output axis for the go (no-go) stimuli and $pop \in \{location, pitch\}$. This value was high when a specific population was strongly modulated by context, i.e. with large activity values along the output axis for its corresponding context and low activity values in the opposite context. We also computed the same ratio for population 0 (considering either contexts as the relevant context) and for randomly labeled neurons. In the latter case, we permuted trial labels for each neuron and relabeled them based on the recomputed pre-stimulus activity with permuted trials. We then used the distribution of output gating calculated on permuted trials to evaluate a permutation test (Fig. 3).

**Communication subspace estimation.** For the multi-area network simulations (see below), we estimated the communication subspaces using Canonical Correlation Analyses (CCA), which is a common approach for aligning neural representations[53–55] and more recently to study multi-area interactions[22,23]. Here, as done previously for studying multi-area interactions[22,23], we focused on noise correlations. Specifically, we started by running 1000 trials of the go/no-go context-dependent task. We then focused on the activity during the stimulus presentation, where the selected stimulus and context information was flowing feedforward and feedback, respectively. We then removed the mean activity of each neuron and stimulus conditions[22,23] and computed the canonical dimensions in the following way. First, to avoid overfitting we reduced the dimensionality of the neural activity collected from both areas using PCA (scikit-learn python package[56]), keeping only the 10 dimensions with the most variability. We then used CCA (scikit-learn python package[56]) to find the canonical dimensions, along which the activity from the two areas were maximally correlated. We did this on one half of the trials and then computed the Pearson correlation with the other half of the trials and repeated this process 250 times (folds). When keeping 10 dimensions, we found that the communication subspace was 2D, as expected. However, we noticed that the number of correlated dimensions was sensitive to the number of principal components that we kept in the preprocessing step (Fig. S4c). We estimated the canonical dimensions either using data from all trials or separating by context.

**Angle between subspaces.** To quantify the alignment between the estimated communication subspaces, we computed the subspace overlap[57]. Specifically, we computed the arccosine of the largest singular value of $B^T\hat{B}$, where $B$ and $\hat{B}$ are the basis defined by the across-area connectivity vectors and the estimated subspace, respectively.

## Recurrent Neural Networks
**Go/no-go context-dependent decision-making task.** We implemented an abstraction of the task in the original publication[13] using the NeuroGym toolbox[28]. Briefly, the input was 4-dimensional, reflecting the pitch and location feature in the rat's experiment, in this case called A and B, and the two contexts, context A and context B. During stimulus presentation, we added gaussian noise with $\sigma = 1$ on top of the

stimuli mean levels. The stimuli features had two levels (-1,1) as well as the contextual inputs (0,1). Before the stimulus presentation, which lasted 10 timesteps, there was a pre-stimulus period of 4 timesteps. Contextual inputs were delivered during both periods, in contrast to the stimuli that were delivered exclusively during stimulus presentation. As was the case for the rats, the network had to select the relevant stimuli level and ignore the irrelevant stimuli, depending on the context level (A=-1, B=* in context A, and A=*, B=-1 in context B).

**Continuous-time RNN.** The dynamics of each unit $i$ were determined by the of recurrent weights $J_{ij}$ and feedforward inputs weights $I_i^l$:

$$\tau \dot{x}_i(t) = -x_i(t) + \frac{1}{N}\sum_j J_{ij}\phi(x_j(t)) + \sum_l^{N_{input}} u^l(t)I_i^l + \eta_i(t) \quad (2)$$

With $\phi = tanh$. The time constant $\tau = 100ms$ was the same for all neurons $i$. For simulation and training, the equation was solved using Euler's method with a time step $\Delta t = 20ms$. The independent white noise term $\eta_i$ was simulated by drawing at each time step from a gaussian with mean 0 and standard deviation 0.05. To calculate the gain of each neuron $\phi'(x_i)$, we passed each neuron activity through $\phi'(x_i) = 1 - tanh(x_i)^2$.

**Trained A1 network.** For the A1 network in isolation, the connectivity matrix $J$ was constrained to be low-rank during training. We found empirically during training that a rank-one network could solve the task, meaning that $J_{ij} = m_i n_j$. The network received $N_{input} = 4$, $(u^A, u^B, u^{xA}, u^{xB})$, corresponding to stimulus $A, B$ and context $A, B$. The input vectors (e.g. $I^A$) defined the sensory axes. We trained the networks using backpropagation through time to minimize the following mean squared error loss function during the last timestep of each trial $t$:

$$L = \sum_t (z_t - X^T w)^2 \quad (3)$$

Where $z_t$ is the correct response on trial $t$, $X$ the network activity during the last timestep and $w$ the readout vector. Only the contextual inputs ($I^{xA}, I^{xB}$) and recurrent weights ($m, n$) were optimized during 64000 trials (in batches of 160 trials each). Optimization was carried out using Adam[58] in pytorch[59] with the decay rates of the first and second moments of 0.9 and 0.999, and learning rate of 0.001.

**Low-rank theory.** We found empirically that rank-one connectivity (i.e. $J = m \otimes n$) was enough to solve the task of interest. It can therefore be shown that the network activity is constrained to be at most $(1 + N_{inputs})$ dimensional:

$$x(t) = \kappa(t)m + \sum_l^{N_{input}} v_l(t)I_l \quad (4)$$

Where $v(t)$ is the low-pass filter version of the input $u(t)$[19]. In this setting, $n$ can be seen as the input-selection vector and $m$ as the output vector of a single latent variable $\kappa$. Previous theoretical work has shown that computations performed by low-rank networks, including those trained through back-propagation, are fully determined by the rank of their connectivity matrix and the geometric relationship of its connectivity vectors - $m, n, I^A, I^B, I^{xA}, I^{xB}$ for the case of the trained A1 network. This relationship is characterized by the overlaps between the different connectivity vectors (e.g. $\sigma_{mn} = m^T n$), which can be subdivided into an arbitrary number of subpopulations[19]. While the rank determines the number of latent variables $\kappa$ that can be manipulated, the number of populations $\mathcal{P}$ constrains the possible computations on the latent variables. For a given rank-one network,

with $\mathcal{P}$ populations, it can be shown that in the limit of $N \to \infty$ the dynamics of the latent variables $\kappa$ is described by:

$$\dot{\kappa} = -\kappa + \sum_p^{\mathcal{P}}\left[\tilde{\sigma}_{nm}^{(p)}\kappa + \sum_i^{N_{input}} \tilde{\sigma}_{nl_i}^{(p)} v_i\right] \quad (5)$$

With $\tilde{\sigma}_{mn}^{(p)} = \sigma_{mn}^{(p)}\langle\phi'\rangle^{(p)}$, which can be seen as the functional connectivity – i.e. a function of the effective connectivity $\sigma_{mn}^{(p)}$ and the population average gain $\langle\phi'\rangle$. Different populations can have different functional connectivity, depending on their average gain, which is itself a recurrent function of $x$ and the active inputs[18,19].

## Inferring populations

To infer the minimal number of populations necessary to solve the task, we followed a previously proposed approach[19]. Briefly, we used the method BayesianGaussianMixture from the scikit-learn python package[56] to cluster neurons in an increasing number of independent populations. After clustering, we calculated the empirical means and covariance matrices of each cluster (i.e. population) independently. We then sampled new connectivity vectors from multivariate gaussian distributions defined by these mean and covariance matrices and concatenated across populations, effectively destroying each neuron identity but keeping the overall correlations. Finally, we evaluated the performance of networks with the sampled connectivity.

**One solution for context-dependent, go/no-go tasks.** We found out that rank-one connectivity with 3 populations solves our task. Neurons in all populations are selective to all external stimuli, but they differ in which stimulus is integrated into the latent variable (or output axis). While the first 2 populations select 1 stimulus (i.e. $\sigma_{nl_A}^{(1)} > 0, \sigma_{nl_B}^{(2)} > 0$) and should ignore the other (i.e. $\sigma_{nl_B}^{(1)} = 0, \sigma_{nl_A}^{(2)} = 0$), the third population must have negative feedback ($\sigma_{nm}^{(3)} < 0$), which we found out to be essential to implement the no-go condition ($\kappa = 0$)[18]. See Fig. S2e,f for the dynamics of this model.

**A1-PFC network.** In contrast to the A1 RNN, in which we trained the connectivity vectors, we directly engineered the A1-PFC RNN. Specifically, we adapted the low-rank framework to describe across-area dynamics. To model multi-area interactions, we represent the connectivity matrix $J$ in terms of a block structure:

$$J = \begin{bmatrix} A = m^A \otimes n^A & J_{P \to A} = I^{P \to A} \otimes n^{P \to A} \\ J_{A \to P} = I^{A \to P} \otimes n^{A \to P} & P = m^P \otimes n^P \end{bmatrix} \quad (6)$$

Recurrent connectivity $A$ and $P$ populates the diagonal and feedforward $J_{A \to P}$ and feedback $J_{A \to P}$ the off diagonal. Our key assumption is that each block has a rank-one structure, and is thus defined by the outer product of two connectivity vectors (e.g. $A = m_A \otimes n_A$). Under this constraint, we can separate the recurrent, feedforward, and feedback inputs in a compact form for the dynamics within A1 and PFC:

$$\dot{x}_i = -x_i + \frac{m_i^A}{N_A}\sum_{j \in A} n_j^A \phi(x_j) + \frac{I_i^{P \to A}}{N_P}\sum_{k \in P} n_k^{P \to A}\phi(x_k) + \sum_l^{N_{input}^A} u_l^A(t)I_i^l, \ i \in A. \quad (7)$$

$$\dot{x}_i = -x_i + \frac{m_i^P}{N_P}\sum_{j \in P} n_j^P \phi(x_j) + \frac{I_i^{A \to P}}{N_A}\sum_{k \in A} n_k^{A \to P}\phi(x_k) + \sum_l^{N_{input}^P} u_l^P(t)I_i^l, \ i \in P. \quad (8)$$

For the purpose of this study, we assume that context information ($u^x$) was delivered only to PFC along $I^x$. Moreover, to ensure a rank-one

feedback communication subspace from PFC to A1, we assume A1 receives an additional constant input $I^k$, the details of which are described at the end of this section. In turn, stimuli $(u^A, u^B)$ are delivered exclusively to A1. Thus, $N_{input}^A = 2$ and $N_{input}^P = 1$. Under these assumptions, in the limit of $N \to \infty$ and assuming again $\mathcal{P}$ populations, the dynamics of the high-dimensional A1-PFC network can be reduced to the dynamics of the following latent variables in A1:

$$\dot{\kappa}_A = -\kappa_A + \sum_p^{\mathcal{P}} \left[ \tilde{\sigma}_{n^A m^A}^{(p)} \kappa_A + \sum_l^{N_{input}} \tilde{\sigma}_{n^A I^l}^{(p)} v^l + \tilde{\sigma}_{n^A I^{P \to A}}^{(p)} v_{P \to A} \right] + \tilde{\sigma}_{n^A I^k} \quad (9)$$

$$\dot{v}_{P \to A} = -v_{P \to A} + \sum_p^{\mathcal{P}} \tilde{\sigma}_{m^P n^{P \to A}}^{(p)} \kappa_P \quad (10)$$

And in PFC:

$$\dot{\kappa}_P = -\kappa_P + \sum_p^{\mathcal{P}} \left[ \tilde{\sigma}_{n^P m^P}^{(p)} \kappa_P + \tilde{\sigma}_{n^P I^x}^{(p)} v_x + \tilde{\sigma}_{n^P I^{A \to P}}^{(p)} v_{A \to P} \right] \quad (11)$$

$$\dot{v}_{A \to P} = -v_{A \to P} + \sum_p^{\mathcal{P}} \tilde{\sigma}_{m^A n^{A \to P}}^{(p)} \kappa_A \quad (12)$$

In addition to the internal latent variables $\kappa_A$, $\kappa_P$, there are now two extra latent variables corresponding to the communication subspace $v_{A \to P}$, $v_{P \to A}$. The key new elements in this formulation are the overlaps (e.g. $\sigma_{n^{A \to P} m}$) between the output vectors within an area, such as m in A1, and the corresponding input-selection vectors populating the off diagonal of J, such as $n^{A \to P}$. Reminiscent of the case for within area dynamics[18–20], non-negative overlap leads to the communication of the corresponding latent variables. Generally speaking, there is one of these overlaps for each within-area variable. In the simplified case addressed here, we set all the overlaps to zero, except for those related to the within-area latent variables. For simplicity, we set $n^{A \to P} = m$ and $n^{P \to A} = m^P$, ensuring both across-area overlaps were non-zero. The overlaps within A1 were set similar to the trained RNN, and when modulated by context, this network integrated the relevant stimuli along the output axis m with the same mechanism as the trained network. In turn, the geometry of PFC was set so it integrated its inputs (i.e. context) into one of the two fixed points. Specifically, we set $\sigma_{mn} > 0$ and $\sigma_{nI} > 0$.

To encode context through a one-dimensional communication subspace in the feedback direction, we introduced a bias term. This bias term, which was fixed and present in all trials and timepoints was defined as $I_k = \frac{I_{ctxA} + I_{ctxB}}{2}$. This way the context value (-1 or 1) was projected along one dimension, which we conveniently defined as $I_x = \frac{I_{ctxA} - I_{ctxB}}{2}$, making the net input $I_{ctxA}$ when context = 1 and $I_{ctxB}$ when context = −1.

### Reporting summary
Further information on research design is available in the Nature Portfolio Reporting Summary linked to this article.

## Data availability
Data are currently available at crcns.org/data-sets/pfc/pfc-1.

## Code availability
The code necessary to replicate all figures are available at https://github.com/jmourabarbosa/multi-area-ctx.

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

## Acknowledgements

We thank Heike Stein, Manuel Molano and Ramon Nogueira for feedback on the manuscript. This work was supported by FrontCog grant ANR-17-EURE-0017, ANR-JCJC-DynaMiC (YB), Institut Universitaire de France (YB), and the NIH Brain Initiative project U01-NS122123 (SO, JB). JB was supported by the Fyssen Foundation.

## Author contributions

J.B., R.P, S.O. and Y.B analyzed the data. J.B. and S.O. conceived and performed the modeling research. C.R. and M.D. performed the experiments. J.B., S.O. and Y.B. wrote the manuscript. All authors critically revised and edited the manuscript.

## Competing interests

The authors declare no competing interests.
