## [Peer Review File · Nature Communications]

Early selection of task-relevant features through population gatingREVIEWER COMMENTS

Reviewer #1 (Remarks to the Author):

In their paper, Barbosa et al. investigate population-level mechanisms for the flexible, context-dependent detection of sounds during an auditory go/nogo task, in which the relevant sound features alternate in blocks. The authors re-analyse neural activity recorded in the primary auditory cortex and medial prefrontal cortex in rats. They find that context is persistently encoded in both A1 and mPFC and that during the relevant blocks, the population response to the go stimulus is enhanced along a selection axis. Using the recently developed framework of low-rank RNNs, they construct an A1 network model that reproduces key aspects of the neural data, such as the context-dependent activity along a selection axis and multiple contextual subpopulations in A1. They extend their model to a two-area setting where low-dimensional A1-PFC interactions might underlie the observed results, with A1 sending selected stimuli to PFC, and PFC feeding back context to A1.

Altogether, the paper addresses an important and open question in the field: how is flexible decision-making implemented in the brain, and how do areas across the sensory-to-decision axis cooperate to achieve this? The paper has two key novelties compared to previous work. First, it proposes three separate subpopulations in A1 that implement population-level gating of inputs, and the authors provide empirical evidence consistent with the hypothesis (but see below). Second, Barbosa et al suggest an RNN model that solves the context-dependent auditory selection task, and is able to explain activity characteristics of both A1 and PFC. The idea that only activity along certain (relevant) axes is communicated, is implemented in a straightforward and clear manner.

That said, the manuscript would benefit from alterations to highlight to the reader which aspects of the hypothesized model are crucial to the performance, which are supported by the data, and which are speculative. In some sense, their model of selective interarea communication seems highly custom designed (the contextual inputs to PFC and specifically the interareal connectivity), and central concepts of the model (e.g., the communication subspace) are based on assumptions, not data. I point out some suggestions for improving the manuscript below. Altogether, however, I find the manuscript is of relevance to many researchers interested in the mechanisms of selective propagation of information between areas in diverse task contexts.

Major comments:

(1) I was surprised to learn in the Methods that all population analyses of the neural data are done on pseudo-population activity. Strictly speaking, the decoding analysis (Figure 1c, 4a) can therefore not be trusted, as it relies on the assumption that there are no noise correlations that could interfere with the decoding. Since this is mostly a modelling paper, that is fine, but it should be clearly stated in the main text!

(2) One of the main predictions of the RNN model is that “different populations select specific stimuli in their preferred context” (line 62). First, it is unclear to me whether the model RNN of A1 (Fig. 2) predicts three discrete populations. Perhaps presenting some more data about the three populations beyond the small block diagrams in Fig 2a would help to characterize this. More importantly, however, it is not clear to me to what extent these three populations actually exist in the data. The analysis performed and presented in Figure 3 will inevitably lead to three populations (those with increased, decreased or similar firing between contexts). However, the initial distribution was unimodal, not trimodal, so that, a priori, there is no evidence for the three populations in the data. So I am not convinced that the predictions of the model match the data. Please clarify or provide some other marker for the existence of three populations. In the discussion, the authors stress that PFC does not show the three populations underlying a selection mechanism. Does this mean that context in PFC is encoded independently of the selection signal coming from A1? Perhaps the authors can highlight the importance of this finding a bit (line 224). The authors can also briefly consider replicating Figure 3a,c for PFC in a supplementary figure.

(3) In the two-area model, PFC acts both as the area that provides the context to A1 (which it itself receives from somewhere else), and as the area that receives the relevant stimulus to guide the choice. The selective inter-area communication through orthogonal channels is a nice feature of the model that allows PFC to receive and feed back these different signals. However, it seems that PFC is rather superfluous and there is no prior reason why these two signals (relaying the selected stimulus and providing context) should be performed by the same area. First, the important context dependent gain modulation occurs in A1. If understood properly, the output from A1 (orange arrows in Fig. 5a) should be sufficient for a downstream motor area to make the right decision. There is no critical computation occurring in PFC (e.g. that depends on the stored contextual information here) that requires PFC to be a relay area. This would be the case if stimulus, choice and outcome are integrated in PFC to infer the context. Second, the contextual inputs to A1 could originate in third area, or alternatively be broadcast to both A1 and PFC. In the model PFC thus acts largely as a routing station: it receives context information and routes it to A1, and it receives selection information from A1 and routes it further downstream. Would a three-area model (a separate decision and context area) perform equally well? The authors should at the very least discuss whether there is a requirement for these two signals to both involve PFC, purely a speculation, or fits with empirical data.

(4) Altogether, the article reads nicely. However, at times the conceptual thread is hard to follow, as it relies too much on jargon stemming from previous work of the authors (e.g., the term ‘subpopulation’, the idea of ‘resampling connectivity’). Unpacking this type of jargon would certainly broaden the appeal of the paper. Furthermore, a conceptual explanation of how a contextual signal could modulate the gain of two/three subpopulations in a one-rank RNN would help the reader understand concepts better, perhaps akin to a figure presented in (<https://adrian-valente.github.io/2022/06/01/low-rank-summary.html>, 9th figure). I also note that Fig. S1a is a very clear figure that conveys many qualitative and quantitative features relevant to the whole manuscript: the ability to decode context throughout the trial, the timing and amplitude of stimulus decoding. In my view, it would benefit the paper to present this in the main figure. For a less invested reader, a direct reproduction of this figure with the connected A1-PFC model could be insightful.

Minor comments:

(1) From the introduction, it is not clear how the author's view of ‘context-dependent tasks’ relates to and differs from other definitions and research: feature-based attention or task switching. It would fit to explicate this relation, especially as the authors stress the importance of their results in terms of relating single neuron gain control to population level mechanisms (and gain control and selective attention have been repeatedly linked in established literature).

(2) Some aspects of the original publication of the data (Rodgers & DeWeese, 2014) are perhaps good to mention briefly. Particularly: (a) Blocks were alternated multiple times (n=?, length?) in a session. (b) Context decoding was not due to drift across the session.

(3) Consider replacing the error bars in from bootstrapped 90% to the more common 95% CI (Figs. 1, 4 and others)

(4) Bootstrapped standard errors of the mean depend on the number of bootstraps (e.g. Figure 3), which is arbitrary. It is suggested to replace bootstrapped SEMs with 95% CIs.

(5) Line 210. The authors consider significant output gating at a p-level of 0.075. If it's a typo, please correct. If not, the authors should rephrase to reflect that gating only significantly occurred in the ‘location’ population.

(6) In Figure 3a the colors appear flipped. In the inset, if the location neurons have higher activity during the location context in blue than pitch context in red, then in the main panel shouldn't the location neurons show a positive modulation of prestimulus activity (measured as location-pitch, header title)?

(7) It could be noted for the reader that Fig. 3c has very different scales for the location and pitch population gating.

(8) Line 341 “could be accounted for by...”

(9) In the Methods it is stated that only correct trials were analysed, but error trial analyses were presented in the original publication (Rodgers & DeWeese, 2014). Could aberrant pre-stimulus contextual coding in a specific subpopulation (only context-A or context-B neurons) underlie errors, or both? Can this be linked to decreased gain for the relevant contextual population?

(10) Figure 5: It could help to give an example connectivity matrix with the within-area and across-area blocks and their features.

(11) Line 482: is this done for each time bin, or an integrated window?

(12) Line 490: what is the time window that is used before stimulus onset?

(13) Line 505: unclear how LDA was used in combination with communication subspace in the multi-area network.

(14) Line 347: Typo: conceivably

Reviewer #2 (Remarks to the Author):

The manuscript by Barbosa et al studies how multi-area interactions may underlie our ability to “attend” to relevant properties of sensory stimuli. To this end, they analyse and model a previously published dataset of (non-simultaneous) PFC and A1 recordings from rats engaged in a context-dependent auditory task. The paper is quite well written, and both the analyses and modelling parts are sound; conclusions are well-supported by the data.

Comments:

1) The description of the task is a bit unclear; for example, how did the rats know whether they were in a pitch or location context? Also, I couldn't find a detailed analysis of the rat's behaviour anywhere in the paper. This should be included both to understand how proficient the rats were —I assume the authors only analysed successful trials?—, and also to figure out whether the smaller effects present in the analysis of neural data during the pitch context (when compared to the location context) may be due to behavioural differences—perhaps the task is just harder for them? (For completeness, some of these “neural differences” are the fact that the trajectories in Figure 1e are less separated than in Figure 1d, and the same is seen in Figure 3c, or that location-location and pitch-pitch look qualitatively different in Suppl Figure 3)

2) Figure 1: (i) Is the selection axis context-independent or does it vary between the pitch and location contexts? (ii) The authors compute the angles between the pitch and location decoding axes, but there may be (many) other decoding axes that provide good accuracy, and these may follow very different alignment profiles. This should be investigated.

3) (i) The authors state that a rank-1 network solves the task. Can you show performance as a function of recurrent rank? (ii) What would happen if the model had a single context input that took one of two values rather than two context inputs?

4) (i) Part of the premise of the paper is that PFC “knows” the context, and several of the authors analyses suggest it has to be this way. But I don't think this is shown explicitly; it should be. (ii) Does PFC encode the stimulus or the choice? Looking at the error trials may help separate which one it is.

5) Figure 3: (i) It would be nice to see Figure 3c for the non-modulated population (grey population in a) -> can one see the separation shown in this panel even if single neurons do not prefer any specific context? (ii) In a similar vein, what do you mean by saying in Line 171 that population 0 reflects global dynamics? Is there a figure that shows this and I may have missed?

6) Figure 4: (i) If one can't decode sensory information from PFC, how is the sensory axis defined in panel b? (ii) This is the authors' paper, not mine, but this figure may be better placed after Figure 1 to present all the experimental data first, and then go on with the modelling part rather than going back and forth.

7) Figure 5: (i) The timing analysis in panel b is very interesting, however it looks as if A1 and mPFC have a similar latency for the location context, which is never discussed.

Specific comments:

- Abstract: (i) spell out the names of the areas before using acronyms; (ii) consider mentioning that you study an auditory task before talking about A1
- Figure 1: I'm not sure two rats help understand the task. Panel a, upper right: some axis labels are confusing. Panel d: how do these angles compare to those of random vectors of the same dimensionality?
- Lines 105-8: expand this sentence to highlight the asymmetry.
- Figure 2d: I found this panel a bit hard to understand.
- Line 210: Aren't the location and pitch populations compared against shuffled data? The text says population 0.
- Figure 5: (i) The central and right hand side parts of the bottom row in panel a are very hard to understand. (ii) In the left hand side: why is the selection axis not orange as it is elsewhere? (iii) there are some arrowheads missing in panel a
- All references to Figure 5 are missing in the text
- Suppl Figure 1b: What are the axes and the points?
- Suppl Figure 2f: what are the two sets of line?
- Suppl Figure 3: Could you clarify why both pitch and location are both decodable from the pitch population in the pitch context?
- Suppl Figure 4, Panel c: (i) Do A1 and PFC mean the angles between m_A and the communication space and m_P and the communication space, respectively? What are the distributions over? (ii) There's a typo on the caption
- In the text around Figure 5, the authors first talk about "developing predictions about the interaction that could be tested in neurophysiological recordings" but then focus on analysing the model; this is a bit confusing.
- You mention using both logistic regression and LDA. Is it correct that all analyses are done using logistic regression, except the projection to the output axis in Fig 3c, which is LDA? If so, why not use logistic regression there, too? Is there a difference between the axes the two methods find?
- This is of course personal, but I found the colour coding (grey versus orange a bit confusing). Perhaps colour adding an additional layer of complexity by changing the palette between the two contexts would help (e.g. using black and red for one context and grey and orange for the other, or something like that).

Reviewer #1 (Remarks to the Author):

In their paper, Barbosa et al. investigate population-level mechanisms for the flexible, context-dependent detection of sounds during an auditory go/nogo task, in which the relevant sound features alternate in blocks. The authors re-analyse neural activity recorded in the primary auditory cortex and medial prefrontal cortex in rats. They find that context is persistently encoded in both A1 and mPFC and that during the relevant blocks, the population response to the go stimulus is enhanced along a selection axis. Using the recently developed framework of low-rank RNNs, they construct an A1 network model that reproduces key aspects of the neural data, such as the context-dependent activity along a selection axis and multiple contextual subpopulations in A1. They extend their model to a two-area setting where low-dimensional A1-PFC interactions might underlie the observed results, with A1 sending selected stimuli to PFC, and PFC feeding back context to A1.

Altogether, the paper addresses an important and open question in the field: how is flexible decision-making implemented in the brain, and how do areas across the sensory-to-decision axis cooperate to achieve this? The paper has two key novelties compared to previous work. First, it proposes three separate subpopulations in A1 that implement population-level gating of inputs, and the authors provide empirical evidence consistent with the hypothesis (but see below). Second, Barbosa et al suggest an RNN model that solves the context-dependent auditory selection task, and is able to explain activity characteristics of both A1 and PFC. The idea that only activity along certain (relevant) axes is communicated, is implemented in a straightforward and clear manner.

That said, the manuscript would benefit from alterations to highlight to the reader which aspects of the hypothesized model are crucial to the performance, which are supported by the data, and which are speculative. In some sense, their model of selective interarea communication seems highly custom designed (the contextual inputs to PFC and specifically the interareal connectivity), and central concepts of the model (e.g., the communication subspace) are based on assumptions, not data. I point out some suggestions for improving the manuscript below. Altogether, however, I find the manuscript is of relevance to many researchers interested in the mechanisms of selective propagation of information between areas in diverse task contexts.

We thank the reviewer for the overall positive feedback and we address the specific comments below.

Major comments:

(1) I was surprised to learn in the Methods that all population analyses of the neural data are done on pseudo-population activity. Strictly speaking, the decoding analysis (Figure 1c, 4a) can therefore not be trusted, as it relies on the assumption that there are no noise correlations that could interfere with the decoding. Since this is mostly a modelling paper, that is fine, but it should be clearly stated in the main text!

We agree with the reviewer that noise correlations could impact our results. We have added a supplementary figure (Supp figure 6) where we repeat our major analyses of Figure 1, 3 and 4 using simultaneously recorded populations of neurons. As expected, the decoding performance is overall lower, but the results are qualitatively very similar. In addition to this figure, we have added two sentences in the methods and results description, respectively:

“Crucially, our results do not depend on these methodological choices and are qualitatively similar when decoding from simultaneously recorded populations (Fig. S6)” (methods)

“In this study we focused on pseudo-trials (Methods) but results were qualitatively similar when analyzing simultaneously recorded neurons (Fig. S6)” (results)

Supplementary Figure 6: Decoding from simultaneously recorded neural ensembles leads to qualitatively similar results as decoding from pseudo-trials. **a**) Size of ensembles recorded from A1 (top) and mPFC (bottom). Most sessions only have 1 recorded neuron, but some have larger size. **b**) Decoding from these small ensembles leads to qualitatively similar decoding results as Fig. S1, albeit with lower accuracy. **c**) Output gating was also qualitatively similar to pseudo-population decoding.

(2) One of the main predictions of the RNN model is that “different populations select specific stimuli in their preferred context” (line 62). First, it is unclear to me whether the model RNN of A1 (Fig. 2) predicts three discrete populations. Perhaps presenting some more data about the three populations beyond the small block diagrams in Fig 2a would help to characterize this.

Evidence for populations in the trained RNNs

We thank the reviewer for pointing out this unclear, but important point. Discrete clusters in the connectivity do not necessarily lead to multiple peaks in the activity histograms. We have strong evidence for the existence of discrete clusters in the connectivity of trained RNN, assessed by shuffling the connectivity structure within each cluster, as we now mention in the main text:

“A key empirical test of this finding is that networks generated by resampling connectivity parameters should solve the task with an accuracy similar to trained ones, as long as the statistics of connectivity within each population are preserved (Dubreuil et al. 2022). Performing this analysis (Inferring populations in Methods), we found that our trained networks relied on at least three populations, as resampling the connectivity vectors from less than three populations did not achieve high performance (Fig. 2a, bottom right).”

This is the main focus of a recent study from the lab, “The role of population structure in computations through neural dynamics”, Nature Neuroscience volume 25, 783–794 (2022), which we cite specifically in this context and is described in detail in the methods section “Inferring populations”. We have

now added a supplementary figure (Sup Fig 7) showing that both in networks optimized with backprop and “hand designed” with discrete populations, the firing rate differences are, like in the data, unimodal.

Supplementary Figure 7: Both in trained (left) and engineered RNNs (right) populations can be identified by their pre-stimulus firing rate, which is higher in opposite contexts (insets). Note that despite a discrete population structure, the distribution of pre-stimulus firing rate difference across contexts is unimodal. Compare this figure to A1 data in Fig. 3

This figure is now referenced when performing similar analyses in A1.

Finally, using our model that we knew had 3 populations, we developed a permutation test (output-gating test, Methods) to test which population had significant context-modulation (Fig S2d) and then applied it to the data (see below).

More importantly, however, it is not clear to me to what extent these three populations actually exist in the data. The analysis performed and presented in Figure 3 will inevitably lead to three populations (those with increased, decreased or similar firing between contexts). However, the initial distribution was unimodal, not trimodal, so that, a priori, there is no evidence for the three populations in the data. So I am not convinced that the predictions of the model match the data. Please clarify or provide some other marker for the existence of three populations.

Evidence for populations in A1, but not in PFC

The reviewer is concerned that because any unimodal distribution can be divided in three populations (those with increased, decreased or similar firing between contexts), this separation is trivial. In addition to the evidence provided above that clusters in the connectivity do not necessarily lead to clusters in the neural activity, we want to point to the important fact that we

are grouping neurons during different time points (pre and during stimulus) and compare them along different variables (context and stimulus). The interesting observation is not that the distribution of spontaneous activity can be divided into three groups, which indeed would be trivial. Instead, we found that A1 neurons grouped by their pre-stimulus selectivity to context show significantly different stimulus-dependent activity during the stimulus. To show that this link is not trivial, we quantified it (Fig 3c) with a permutation test. This test, which was validated in the model (Fig S2d), shows that neurons grouped randomly (i.e. permuted) do not show as much context-modulation as unpermuted neurons:

This link between context modulation and stimulus-related dynamics is not trivial, as we grouped neurons during different time points (before and during stimulus) and compared them along different variables (context and stimulus). As done with the simulations, we quantified the level of context-dependent population dynamics (output gating, Methods) and compared it with randomly selected populations.

Finally, when using the same approach in PFC activity, we don't find that neurons grouped by their pre-stimulus activity are differentially modulated in opposite contexts, showing that our analyses do not necessarily lead to three different clusters. See also the answer to the question immediately below.

In the discussion, the authors stress that PFC does not show the three populations underlying a selection mechanism. Does this mean that context in PFC is encoded independently of the selection signal coming from A1? Perhaps the authors can highlight the importance of this finding a bit (line 224).

What makes context and selection to be encoded independently is that each variable is encoded along orthogonal directions within the state space (Figure 5a, bottom right). In the model, this can occur regardless of the population structure: while A1 has 3 populations and PFC just one, both encode the selected stimulus and context in orthogonal dimensions. The complementary role of populations and dimensionality is not the focus of our paper, but it is the main focus of Dubreuil et al.

Instead, what we meant in the Discussion was that PFC, in contrast to A1 (Fig 3d), does not pass our output-gating permutation test. This test, as described in the methods, tests if the populations selected by their pre-stimulus firing

rate have different dynamics in each context and, more importantly, if that difference is higher than expected by grouping neurons randomly. Our interpretation, that we incorporate in our model, is that PFC neurons are well described by one population (in contrast to A1, which needs at least 3).

We have modified our results description to clarify our interpretation and modelling:

PFC results:

In sum, PFC neurons encode context and decision along orthogonal directions. In contrast to A1, they do not encode the irrelevant stimulus and their activity seems to be well described by one population. Next, we incorporate these observations in a multi-area network.

Model section

Specifically, we set the connectivity geometry of A1 similarly to the trained network (Fig. 2), meaning that when biased by contextual inputs it selected the relevant stimuli along the A1 output axis through population gating (Fig. 2,3). In turn, PFC was set up to store the current context in persistent activity using only one population, as suggested by the data analyses.

The authors can also briefly consider replicating Figure 3a,c for PFC in a supplementary figure.

We added a new figure (Fig. S8 as per reviewer suggestion), including all populations from both areas.

Supplementary Figure 8: Similar analyses performed in Fig. 3c for the two main population in A1, here done for all populations of A1 and PFC. See also Fig. 2a

(3) In the two-area model, PFC acts both as the area that provides the context to A1 (which it itself receives from somewhere else), and as the area that receives the relevant stimulus to guide the choice. The selective inter-area communication through orthogonal channels is a nice feature of the model that allows PFC to receive and feed back these different signals. However, it seems that PFC is rather superfluous and there is no prior reason why these two signals (relaying the selected stimulus and providing context) should be performed by the same area.

First, the important context dependent gain modulation occurs in A1. If understood properly, the output from A1 (orange arrows in Fig. 5a) should be sufficient for a downstream motor area to make the right decision. There is no critical computation occurring in PFC (e.g. that depends on the stored contextual information here) that requires PFC to be a relay area. This would be the case if stimulus, choice and outcome are integrated in PFC to infer the context.

Second, the contextual inputs to A1 could originate in third area, or alternatively be broadcast to both A1 and PFC. In the model PFC thus acts largely as a routing station: it receives context information and routes it to A1, and it receives selection information from A1 and routes it further downstream. Would a three-area model (a separate decision and context

area) perform equally well? The authors should at the very least discuss whether there is a requirement for these two signals to both involve PFC, purely a speculation, or fits with empirical data.

We agree with the reviewer that the integration of stimulus, choice and outcome in PFC to infer the context would be more realistic. We now make it explicit that this case was not considered here but it should motivate for future work:

In our model, there is a division of labor: A1 represents all stimuli, but when biased by the current context it selects the relevant stimulus; in turn, PFC reads out the relevant stimulus and uses it to effectively guide behavior and could in principle infer the current context through trial and error. We leave for future work the question of how the current context is inferred.

Note that right after we mention another more realistic case not addressed here, including the case of a third area:

The computational advantages of our model modularity are unclear when considering a single task, as was the focus of this work. Considering instead a more realistic case, in which rats are trained on more than one task, for instance two context-dependent tasks of different modalities (e.g. auditory and visual), the advantages become apparent: in principle, PFC could be reused in both tasks, storing the current context and biasing currently relevant networks. Similarly, A1 could conceivably route different stimuli to another area, instead of only the relevant stimulus to PFC. In this work, we opted to model the simplest hypothesis that could explain our findings within A1 and PFC.

Given all this, we want to stress that the PFC role in our model is not superfluous. Instead, it acts as a controller of sensory areas, which is a popular hypothesis that has been proposed by Cohen Miller and Cohen 2001 and is still regarded as the dominant hypothesis, as referenced in an earlier review by Okazawa & Kiani 2023 (their Fig 5). We now make it explicit that our model is a network implementation of this conceptual model:

...PFC acts as a controller of A1, dynamically selecting the appropriate communication subspace for the ongoing context. Our model is an explicit implementation of the Miller and Cohen model of PFC (ref. but see Okazawa & Kiani 2023 for a recent review) and complements a

large body of computational work focusing on multi-area interactions (see Perich and Rajan 2020 for a recent review).

(4) Altogether, the article reads nicely. However, at times the conceptual thread is hard to follow, as it relies too much on jargon stemming from previous work of the authors (e.g., the term ‘subpopulation’, the idea of ‘resampling connectivity’). Unpacking this type of jargon would certainly broaden the appeal of the paper.

In the revised manuscript, we have reduced the amount of technical jargon, in particular relating to the specific comments of the reviewer.

Subpopulations

We removed that term from Figure 3 title and changed it to:

Pre-stimulus context-dependent activity reveals three distinct populations

Similarly in the methods:

Identification of distinct populations by pre-stimulus modulation.

We decided to keep one instance of “subpopulation” in the methods when describing the model.

This relationship is characterized by the overlaps between the different connectivity vectors (...) which can be subdivided into an arbitrary number of subpopulations (cite{Dubreuil2022}).

This is because we mention how the connectivity vectors can be subdivided into an arbitrary number of parts, thus the term “subpopulations” seems appropriate.

Resampling connectivity

We have explained the concept of “resampled connectivity” from another angle. In addition to sampling from a distribution, this step can be seen as shuffling the connectivity weights and preserving only the correlations within each population.

In the main document:

A key empirical test of this finding is that networks with shuffled connectivity parameters should still solve the task with an accuracy similar to trained ones, as long as the statistics of connectivity within each population are preserved (\cite{Dubreuil2022}).

In Figure 2 caption:

Right, clustering and shuffling the connectivity (Inferring populations in Methods) keeping an increasing number of populations shows at least 3 populations (population A, B and 0)

An in the methods:

We then sampled new connectivity vectors from multivariate gaussian distributions defined by these mean and covariance matrices and concatenated across populations, effectively destroying each neuron identity but keeping the overall correlations.

Furthermore, a conceptual explanation of how a contextual signal could modulate the gain of two/three subpopulations in a one-rank RNN would help the read understand concepts better, perhaps akin to a figure presented in (<https://adrian-valente.github.io/2022/06/01/low-rank-summary.html>, 9th figure).

We tried to clarify this point in figure 2d, which has a new caption

And is cited in the main text:

“As found in a recent study (Dubreuil2022), context-dependent modulation at the population level relies on selective gain modulation at the single-neuron level. This gain modulation is determined by the working point of single neurons, with neurons with higher firing rate exhibiting lower gain (Fig. 2d, left). For neurons in population B, we calculated their gain as the slope of the transfer function before the stimulus presentation (Fig. 2d, middle Methods) and the corresponding firing rate (Fig. 2d, right) in each context.”

We thank the reviewer for suggesting to improve our explanation.

I also note that Fig. S1a is a very clear figure that conveys many qualitative and quantitative features relevant to the whole manuscript: the ability to decode context throughout the trial, the timing and amplitude of stimulus decoding. In my view, it would benefit the paper to present this in the main figure. For a less invested reader, a direct reproduction of this figure with the connected A1-PFC model could be insightful.

We thank the reviewer for making this suggestion. In the current version of the manuscript, parts of Fig S1 are now in the main manuscript (see Fig 5). As per reviewer suggestion, we have highlighted the timing differences of decoding, which is explained by the model (as can be seen in Fig 5). We opted to not add context decoding to the main figures for two reasons. First, this result was already present in the original work of Rodgers & DeWeese. Second, we believe that adding another line to the plots would blur the main, timing difference result.

Minor comments:

(1) From the introduction, it is not clear how the author's view of 'context-dependent tasks' relates to and differs from other definitions and research: feature-based attention or task switching. It would fit to explicate this relation, especially as the authors stress the importance of their results in terms of relating single neuron gain control to population level mechanisms (and gain control and selective attention have been repeatedly linked in established literature).

We have now added more information about context-dependent tasks to the introduction and discussion. We thank the reviewer for this suggestion, it indeed improves the scope of our results.

Introduction

Empirical evidence demonstrates however that primary sensory areas are modulated by behavioral context (Hajnal2021, Maunsell2006, Paneri2017, Rodgers2014, Siegel2015), potentially through feedback interactions with downstream areas that could control the selection of the relevant stimulus upstream (Fritz2010, Winkowski2013). This evidence supports early models of parallel distributed processing (cohen1990), that proposed that task-relevant stimuli encoding could be enhanced by top-down inputs to sensory neurons. The prefrontal cortex (miller2001) is deemed essential in providing these inputs, which push task-irrelevant units to the low-gain region of their dynamic range, thereby effectively reducing their sensitivity.

Discussion

Our major contribution is to propose a single-neuron mechanism (i.e. gain modulation) for flexible selection of different subspaces through population neural dynamics (Javadzadeh2022, Panichello2021, Yoo2020), despite fixed connectivity. We have focused on feature-selection (Rodgers2014), a specific kind of context-dependent tasks (okazawa2023), and future work will be necessary to extend our results to other types of context-dependent task, such as fear conditioning extinction (heald202, Jercog2021) or task switching (Reinert goltstein 2021, heald2023, okazawa2023).

(2) Some aspects of the original publication of the data (Rodgers & DeWeese, 2014) are perhaps good to mention briefly. Particularly: (a) Blocks were

alternated multiple times (n=?, length?) in a session. (b) Context decoding was not due to drift across the session.

We have now added the required info to the methods:

The task alternated between blocks of localization and pitch discrimination trials. Each localization block began with 20 "cue trials" in which only the localization stimulus played, followed by 60 trials on which both localization and pitch discrimination stimuli played simultaneously. Pitch discrimination blocks similarly began with 20 cue trials of pitch stimuli, followed by 60 trials of both stimuli simultaneously. Localization and pitch discrimination blocks alternated throughout the entire session. The session lasted for approximately one hour, and rats were allowed to do as many trials as they wished during this period of time.

(...)

In the original study (Rodgers2014), we considered the possibility that a difference in encoding between the blocks might arise from a slow firing rate drift over the entire session. To test for this, we fit a linear model that explained each neuron's response as a combination of block number within the session as well as block identity (localization or pitch discrimination). Only a small minority of neurons (3.5%, 8/231) showed an apparent difference between the blocks that was explained by a significant effect of block number and not block identity.

(3) Consider replacing the error bars in from bootstrapped 90% to the more common 95% CI (Figs. 1, 4 and others)

Done.

(4) Bootstrapped standard errors of the mean depend on the number of bootstraps (e.g. Figure 3), which is arbitrary. It is suggested to replace bootstrapped SEMs with 95% CIs.

We want to clarify that we did not calculate the standard error (standard deviation / \sqrt{N}) of the bootstraps, which would indeed depend on the number (N) of bootstraps. Instead, we calculated the standard deviation of the bootstraps, which does not depend on the bootstrap number (orange vs green, below) and, in the case of the gaussian distributions, it is equivalent to the parametric SEM (blue), as can be seen in the simulations shown in figure below.

(notebook to replicate simulations:

<https://colab.research.google.com/drive/135kMbDRllc1uY9hgu0SBAL0JyxhB ynz?usp=sharing>)

This method of calculating the SEM is described in “3. THE BOOTSTRAP” in “Nonparametric estimates of standard error: The jackknife, the bootstrap and other methods”, *Biometrika* (1981), which we now cite to avoid further misunderstandings.

(5) Line 210. The authors consider significant output gating at a p-level of 0.075. If it’s a typo, please correct. If not, the authors should rephrase to reflect that gating only significantly occurred in the ‘location’ population.

The reviewer is right that this sentence is sloppy. We have rewritten it to be more precise:

We found that the output gating of both populations ($p=0.03$ and $p=0.004$, location and pitch populations, respectively), but not population 0 ($p>0.25$) was significantly higher than in randomly selected populations (Fig. Fig3d, hyperref[sec:popanal]{Methods}). Moreover, the location population output-gating strength was significantly higher than population 0 ($p<0.0025$) and close to the typical significance threshold for the pitch population ($p<0.075$).

Note that in this case we could have done a one-sided test, as our model implies a clear direction.

(6) In Figure 3a the colors appear flipped. In the inset, if the location neurons have higher activity during the location context in blue than pitch context in

red, then in the main panel shouldn't the location neurons show a positive modulation of prestimulus activity (measured as location-pitch, header title)?

Thank you, the colours (the x labels, actually) of the inset were indeed flipped but are now corrected.

(7) It could be noted for the reader that Fig. 3c has very different scales for the location and pitch population gating.

We have a paragraph in the supplementary notes speculating about why we think pitch is less decodable, called "*Slight asymmetry between contexts*". To make sure a distracted reader does not miss that note, we are now referencing that note in the caption of Figure 1 and Figure 3.

(8) Line 341 "could be accounted for by..."

Corrected.

(9) In the Methods it is stated that only correct trials were analysed, but error trial analyses were presented in the original publication (Rodgers & DeWeese, 2014). Could aberrant pre-stimulus contextual coding in a specific subpopulation (only context-A or context-B neurons) underlie errors, or both? Can this be linked to decreased gain for the relevant contextual population?

We thank the reviewer for this wonderful idea! Indeed we only analysed correct trials, but given this excellent suggestion we looked into error trials too.

Unfortunately, there were not many usable error trials in general - by usable we mean errors in which the animal actually made a response. In fact, there were many sessions without error trials at all (total of 59 cells recorded in sessions with error trials vs 130 cells during sessions with correct trials). Accidentally, most error trials occurred during sessions where population 0 neurons were recorded and there were no error trials at all in sessions in which neurons from population B were recorded. The population labelling was done using correct trials, as in the main manuscript.

This unfortunate fact (i.e. no data for population B neurons) makes the suggested analyses impossible, but we will keep this in mind for analyses currently being performed in the lab on larger datasets. Note however that we were able to use error trails for another analyses suggested by reviewer #2, that we copy here for easier inspection:

Response to reviewer #2:

We agree it could be interesting to distinguish between stimulus and choice, especially for PFC where choice signals are usually strong. Following reviewer's suggestion, we use error trials to dissociate between the two possibilities. The ideal analyses to clarify this issue would require relevant stimulus and decision completely uncorrelated. Given the structure of the task, in which correct decision was 100% correlated with relevant stimulus, one way of decorrelating both variables would be to have an equal amount of error and correct trials. This makes the animal decision and stimulus uncorrelated and therefore decoding of the relevant stimulus would be isolated from decoding of decision. We have found that it is possible to decode both relevant stimuli and the choice from PFC, even when they are artificially uncorrelated by introducing errors. For completeness, we also perform the same analyses in A1, where relevant stimulus was strongly encoded above and beyond choice signals.

(10) Figure 5: It could help to give an example connectivity matrix with the within-area and across-area blocks and their features.

Following this suggestion, we have a panel to Fig 5a (middle, bottom).

We think this schematics is more informative than the actual connectivity matrix which is not very informative of its specific structure, even when thresholded.

(11) Line 522 (482): is this done for each time bin, or an integrated window?

We rephrase the text to make it more clear that we used an integrated window:

To estimate single-neuron selectivity, we counted all the single-neuron spikes while the stimulus was presented (250 ms) and regressed them against a linear combination of all task variables of interest.

(12) Line 529 490: what is the time window that is used before stimulus onset?

We now added the specific time period that was used:

To test the RNN prediction laid out in Fig. 2, we averaged each neuron's firing rate before stimulus (1 s) onset separately in each context.

(13) Line 547 505: unclear how LDA was used in combination with communication subspace in the multi-area network.

Indeed that was wrong, thank you. In this version of the manuscript we have removed that sentence.

(14) Line 347: Typo: conceivably

Corrected, thank you.

Reviewer #2 (Remarks to the Author):

The manuscript by Barbosa et al studies how multi-area interactions may underlie our ability to “attend” to relevant properties of sensory stimuli. To this end, they analyse and model a previously published dataset of (non-simultaneous) PFC and A1 recordings from rats engaged in a context-dependent auditory task. The paper is quite well written, and both the analyses and modelling parts are sound; conclusions are well-supported by the data.

We thank the reviewer for the positive comments and the constructive recommendations that we address specifically below.

Comments:

1) The description of the task is a bit unclear; for example, how did the rats know whether they were in a pitch or location context?

Indeed this detail was missing. We have now added a sentence that was only described this in the methods to the main text:

Before contextual block changes, the rats performed 20 “cue trials”, in which the rat heard only relevant sounds without the irrelevant feature.

In addition, we expanded the methods description:

The task alternated between blocks of localization and pitch discrimination trials. Each localization block began with 20 “cue trials” in which only the localization stimulus played, followed by 60 trials on which both localization and pitch discrimination stimuli played simultaneously. Pitch discrimination blocks similarly began with 20 cue trials of pitch stimuli, followed by 60 trials of both stimuli simultaneously. Localization and pitch discrimination blocks alternated throughout the entire session. The session lasted for approximately one hour, and rats were allowed to do as many trials as they wished during this period of time.

Also, I couldn’t find a detailed analysis of the rat’s behaviour anywhere in the paper.

We want to clarify that we have already reported a detailed behavioural analysis in another study (Rodgers and DeWeese, 2014). We have now added the following sentence in the first paragraph of the result to make this clear:

For a detailed description of the training procedure as well as behavioral analyses after training, we refer the reader to the original publication (Rodgers and DeWeese 2014).

This should be included both to understand how proficient the rats were —I assume the authors only analysed successful trials?—, and also to figure out whether the smaller effects present in the analysis of neural data during the pitch context (when compared to the location context) may be due to behavioural differences—perhaps the task is just harder for them? (For completeness, some of these “neural differences” are the fact that the trajectories in Figure 1e are less separated than in Figure 1d, and the same is seen in Figure 3c, or that location-location and pitch-pitch look qualitatively different in Suppl Figure 3)

We have a paragraph in the supplementary notes where we speculate along the same lines why we think pitch is less decodable. “*Slight asymmetry between contexts*”. We now reference that note in the caption of Figure 1 and Figure 3.

2) Figure 1: (i) Is the selection axis context-independent or does it vary between the pitch and location contexts?

To maximise the decoding, we have performed the analyses separated by each context, which lead to decoding axes orthogonal across context. However, because the no-go condition was present in both contexts, a context-independent axis can be found by computing the difference between both decoding axes, which results in a go right-go left axis. Here are the figures when the data is projected on such a common axis:

Note that because we are now forcing the selected stimulus to explore the same axis on both contexts, each context will explore a different side of that axis. Here we chose the location context to explore the positive part of the selection axis.

And here the analyses reported in the main document for easier comparison:

As expected the decoding strength (length of the trajectory) is weaker when projecting on a common axis. We have now added this new figure to the supplementary figure 1, with its corresponding caption and new reference in the main text:

To maximize the separation between go and no-go trials, we performed the decoding analyses separately for each context, but a common selection axis across context could be found with similar dynamics (Fig. S1d).

(ii) The authors compute the angles between the pitch and location decoding axes, but there may be (many) other decoding axes that provide good accuracy, and these may follow very different alignment profiles. This should be investigated.

To clarify, we are computing the angle between decoding axis of the same stimuli, when it was irrelevant and relevant, as is now written explicitly in the text:

We quantified the angle between relevant and irrelevant decoding axes and found that they were aligned, but not parallel, as expected in a selection code.

This is not the angle between different stimuli. While we agree that we could find many other axes that could successfully decode go vs no, the axes that we found are the optimal decoding axes - ie. the ones that maximise the distances between conditions.

3) (i) The authors state that a rank-1 network solves the task. Can you show performance as a function of recurrent rank? (ii) What would happen if the model had a single context input that took one of two values rather than two context inputs?

i) We have now added a new figure where we show that allowing for higher rank connectivity matrices, even full rank, does not improve performance. There is however an improvement in training speed when allowing for higher ranks, but full rank networks tend to overfit. Since this is not the main focus of the manuscript, we did not want to explore this question further.

ii) We want to note that both solutions are mathematically equivalent. In our model of Fig5 we show the exact same model (3 populations, 2 with gain modulation) with 1D contextual input. We have also added a new panel where we retrained networks constraining the contextual input to be 1D and a very similar picture to i) emerges.

Supplementary Figure 9: A rank-1 RNN is enough to solve our context-dependent task. Allowing for higher ranks, even full rank, does not improve performance (rightmost column), but higher rank networks learn faster and full rank networks tend to over-fit. Top, network training is the same as in the main manuscript. Bottom, similar to top but contextual input is 1D (as in Fig. 5) instead of 2D.

4) (i) Part of the premise of the paper is that PFC “knows” the context, and several of the authors analyses suggest it has to be this way. But I don’t think this is shown explicitly; it should be.

The reviewer is correct that the paper assumes that the context is “known” to PFC (and A1). We show this explicitly in Fig S1a. The reason why we did not add context decoding to the main figures was because this was the main result of the original work of Rodgers & DeWeese. We have now added to the paragraph describing PFC analyses two sentence explicitly stating the PFC encoded context

...when their activity was strongly selective to context (Fig. S1)...

In sum, PFC neurons encode context (Fig. S1) and decision (Fig. 4), but..

(ii) Does PFC encode the stimulus or the choice? Looking at the error trials may help separate which one it is.

We agree it could be interesting to distinguish between stimulus and choice, especially for PFC where choice signals are usually strong. Following reviewer's suggestion, we use error trials to dissociate between the two possibilities. The ideal analyses to clarify this issue would require relevant stimulus and decision completely uncorrelated. Given the structure of the task, in which correct decision was 100% correlated with relevant stimulus, one way of decorrelating both variables would be to have an equal amount of error and correct trials. This makes the animal decision and stimulus uncorrelated and therefore decoding of the relevant stimulus would be isolated from decoding of decision. We have found that it is possible to decode both relevant stimuli and the choice from PFC, even when they are artificially uncorrelated by introducing errors. For completeness, we also perform the same analyses in A1, where relevant stimulus was strongly encoded above and beyond choice signals.

We have now added this plot to Supplementary Fig. 5 and cite it when discussing motor confounds in the note:

Second, we artificially decorrelated decisions and relevant stimuli by enforcing an equal amount of error and correct pseudo-trials during sampling. We have found that it is possible to decode both relevant stimuli and the choice from A1 and PFC.

And in the main text, when describing PFC analyses:

Notably, decorrelating the animal's choice and the relevant stimuli with error trials, we found that both PFC and A1 encoded the relevant stimulus above and beyond choice (Fig.S5c).

And the model predicting regarding different lag in A1 and PFC:

Due to this model architecture, relevant stimuli are encoded earlier in A1 than in PFC (Fig.5b, left), as seen in the data (Fig.5b), even when artificially decorrelating decisions and relevant stimuli with error trials (Fig.S5c).

5) Figure 3: (i) It would be nice to see Figure 3c for the non-modulated population (grey population in a) -> can one see the separation shown in this panel even if single neurons do not prefer any specific context?

We have now added a new figure (figS9) with all the populations within A1 and PFC. It can be seen that pop 0 has a very different profile than each of the other populations. More importantly, its dynamics are not different from selecting neurons at chance ($p > 0.25$, Figure 3d).

(ii) In a similar vein, what do you mean by saying in Line 171 that population 0 reflects global dynamics? Is there a figure that shows this and I may have missed?

We agree this sentence was a bit obscure. What we mean by global dynamics is that this population exhibits dynamics that are not different from the dynamics of a population selected at chance (output-gating test, Fig S2d). We have now rephrase this sentence to be more clear:

We found that populations A and B showed substantially more context-dependent activity along the readout axis than population 0,

whose dynamics were not different from a randomly selected population of neurons (Fig. S2d).

Relevant part of figure S2 copied here for your convenience:

6) Figure 4: (i) If one can't decode sensory information from PFC, how is the sensory axis defined in panel b?

The axis is defined in the same way as in the case of A1. As expected from the weak decoding performance on Fig 4a, the activity along the sensory axis is very weak. We tried to clarify this when citing that panel:

We also visualized mPFC context-dependent dynamics along sensory and selection axes, estimated similarly to A1 (Fig. 1; Population decoding in Methods). In contrast to A1, mPFC dynamics evolved exclusively along the selection axis encoding the decision and only very weakly along the sensory axis during the irrelevant context (Fig. 4b).

(ii) This is the authors' paper, not mine, but this figure may be better placed after Figure 1 to present all the experimental data first, and then go on with the modelling part rather than going back and forth.

We very much understand the reviewers suggestion. The structure of the paper has been the centre of very long discussions. The order suggested by the reviewer was in fact the one chosen in the first draft. But after going back and forth during the manuscript development, the authors agreed that A1 should be given a special highlight before moving on to PFC and then

combine both in a final model. We appreciate the reviewer's suggestion, but after another discussion we have decided to keep the current structure, which we believe highlights A1 analyses. Note that we have now changed the title slightly to highlight A1 even earlier:

Early selection of task-relevant features through population gating

7) Figure 5: (i) The timing analysis in panel b is very interesting, however it looks as if A1 and mPFC have a similar latency for the location context, which is never discussed.

We thank the reviewer for noting that this was unclear. We have redone the analyses with smaller bins (25ms instead of 50ms) and now show explicitly what bins are significantly different from chance. It is now clear that in both contexts PFC is lagging behind A1 by 2 bins.

Specific comments:

- Abstract: (i) spell out the names of the areas before using acronyms; (ii) consider mentioning that you study an auditory task before talking about A1
Done.

Figure 1: I'm not sure two rats help understand the task.

We think the case of different responses (each rat) for the same stimuli is a key component of our task, i.e. what makes it context-dependent. Unless the reviewer thinks that having the two rats makes the explanation more confusing, we prefer to leave them.

Panel a, upper right: some axis labels are confusing.

Indeed this panel was very crowded. We have removed some of the labels that were redundant with the text, keeping only the important ones.

Panel d: how do these angles compare to those of random vectors of the same dimensionality?

We have added the distribution of the angles of random vectors for comparison. Thanks for this suggestion.

And the caption now reads:

On the left, the angles between sensory and relevant axis before orthogonalization, estimated during location blocks are shown in dark gray; for comparison, angles between random vectors (computed by shuffling the weights of each neuron) are shown in light gray.

- Lines 105-8: expand this sentence to highlight the asymmetry.

We thank the reviewer for pointing out that the asymmetry was not very clear. We have now added a new panel to the first supplementary figure highlighting this asymmetry by providing two extra possible scenarios, the *symmetric* case (directly related with the reviewer's comment) and an additional one, the *extended* scenario.

Supplementary Figure 1: *A1 and PFC encode different task variables.* **a)** Logistic regression decoding (Methods) of location (left), pitch (right) in each context (red and blue) and decoding of context overlaid on both plots for comparison. Top, A1 encodes both stimulus' features in either contexts. Bottom, PFC encodes only the relevant stimulus' features for the ongoing context. **b)** Left, feature-selectivity is mixed in both areas. Each axis corresponds to the beta weights corresponding to location and pitch selectivity of the linear model (Methods). Each dot is a neuron. Right, fraction of cells with significant task-variables regressors (Methods). Error-bars are bootstrapped SEM (Efron 1981). **c)** The three possible scenarios detailed in the main figure do not provide an exhaustive representation of all possible scenarios. Here we illustrate two other possible scenarios. In the extended scenario, the relevant stimulus is enhanced (similarly to the selected code), but both axes are parallel, so a selection axis does not exist. The symmetrical case is very similar to the selected code. In both scenarios there is a selection axis, but in the selected code the relevant go stimulus encoding is further enhanced. The important message is that the angle between relevant and irrelevant decoding axis and the decoding performance fully characterize the encoding geometry of A1. **d)** Similar analyses to Fig. 1, but forcing the activity during both context to explore the same selection axis. This common axis was determined as the average each context selection axis.

And highlighted it in the main text:

Note that the three scenarios detailed here do not provide an exhaustive list of all the possible encoding geometries (see Fig. 1c for two other scenarios). Importantly however, each encoding geometry is fully characterized by the angle between relevant and irrelevant axes and their across-context decoding performance (Fig. 1b).

- Figure 2d: I found this panel a bit hard to understand.

We reorder the different panels and their explanation to make them easier to understand

And is explained accordingly in the main text:

“As found in a recent study (Dubreuil2022), context-dependent modulation at the population level relies on selective gain modulation at the single-neuron level. This gain modulation is determined by the working point of single neurons, with neurons with higher firing rate exhibiting lower gain (Fig. 2d, left). For neurons in population B, we calculated their gain as the slope of the transfer function before the stimulus presentation (Fig. 2d, middle Methods) and the corresponding firing rate (Fig. 2d, right) in each context.”

- Line 210: Aren't the location and pitch populations compared against shuffled data? The text says population 0.

The original text was a bit confusing. A new sentence now clarifies what we did:

We found that the output gating of both populations ($p=0.03$ and $p=0.004$, location and pitch populations, respectively), but not population 0 ($p>0.25$) was significantly higher than in randomly selected populations (Fig. Fig3d, \hyperref[sec:popanal]{Methods}). Moreover, the location population output-gating strength was significantly higher than population 0 ($p<0.0025$) and close to the typical significance threshold for the pitch population ($p<0.075$).

- Figure 5: (i) The central and right hand side parts of the bottom row in panel a are very hard to understand.

Thank you for pointing that out. We have rewritten the caption explaining these panels trying to clarify our analyses and interpretation:

Highlighted in gray is the first canonical dimension of each context, used in the remaining panels. In the middle, we show that the first canonical dimensions of context A and B (left) are orthogonal, i.e. that A1 and PFC communicate through orthogonal subspaces in opposite contexts (red and blue, Methods), despite fixed connectivity. However, we show that these orthogonal subspaces are aligned along the output axis (m^A , purple). On the right, we show that these orthogonal subspaces are supported by different populations. Neurons of

population A have mostly null coefficients on the communication subspace during context B; conversely for neurons in population B.

(ii) In the left hand side: why is the selection axis not orange as it is elsewhere?

In our framework, there are actually two selection vectors, one within A1 and another within PFC. The purple vector, within A1, selects the relevant stimulus but it is the orange vector, within PFC, that projects the relevant stimulus information into the activity of PFC.

(iii) there are some arrowheads missing in panel a

We think the reviewer is referring to the black dots between areas. These dots represent the latent variables that are shared across areas. We have now explicitly mention that in the caption to avoid confusion:

(...) Solid black circles represent the two variables that are shared across areas. (...)

- All references to Figure 5 are missing in the text

Thank you, corrected now.

- Suppl Figure 1b: What are the axes and the points?

Thanks for pointing to these unclear descriptions. We have now modified the caption, which reads:

Left, feature-selectivity is mixed in both areas. Each axis corresponds to the beta weights corresponding to location and pitch selectivity of the linear model (Methods). Each dot is a neuron.

- Suppl Figure 2f: what are the two sets of line?

We have added a new sentence to clarify what the colored lines are:

Here red, gray and blue correspond to "go left", "no go" and "go right", respectively.

- Suppl Figure 3: Could you clarify why both pitch and location are both decodable from the pitch population in the pitch context?

We need to clarify that this is the axis of the relevant stimulus (see Fig 1). While it is true that we can also decode the irrelevant stimulus, this is visible only along another axis (sensory axis, Fig 1). This is the focus of Fig. 1. This supplementary figure is just to show that indeed there is context modulation on each population and it is not just an artefact of averaging different conditions in each context. What the reviewer might be noticing is that pitch is not as decodable as location and we have now highlighted this in the caption:

As for other analyses, pitch decoding is weaker than location (note the different y-axis).

- Suppl Figure 4, Panel c: (i) Do A1 and PFC mean the angles between m_A and the communication space and m_P and the communication space, respectively? What are the distributions over?

We have changed the caption slightly to clarify this:

In red and blue, the bootstrapped angle between the subspaces estimated with canonical correlation analysis and those defined by the network connectivity (Fig. 5a).

(ii) There's a typo on the caption

Corrected, thank you.

- In the text around Figure 5, the authors first talk about “developing predictions about the interaction that could be tested in neurophysiological recordings” but then focus on analysing the model; this is a bit confusing.

Thank you, we have now changed the text to make it clear that testing the predictions is beyond the scope of the current manuscript (because we don't have simultaneous recordings):

We then developed predictions about the interaction between A1 and PFC that could potentially be tested in datasets with simultaneously recorded areas.

- You mention using both logistic regression and LDA. Is it correct that all analyses are done using logistic regression, except the projection to the output axis in Fig 3c, which is LDA? If so, why not use logistic regression there, too? Is there a difference between the axes the two methods find?

We have used logic regression when we wanted a measure of accuracy, such as in most decoding figures as the reviewer has rightly pointed out. In Fig 3 we are interested in quantifying how well go vs no-go conditions are separated (not accuracy, which saturates at 100%). So for this analysis we calculated directly the distances between the classes. We call this LDA because in LDA one also computes the distance between classes. Strictly speaking, these distances are also normalised by their covariances, but since we are using pseudo populations the normalisation step should not have any effect. After the reviewer's comment, we realised that this made the analysis description unclear. We have described our analysis in more detail instead of just calling it LDA:

To estimate the output axis in Fig. 3, we computed the distance between the average activity during go and no-go condition (Bagur2018). We then projected the activity separated for each condition along the same axis.

- This is of course personal, but I found the colour coding (grey versus orange a bit confusing). Perhaps colour adding an additional layer of complexity by changing the palette between the two contexts would help (e.g. using black and red for one context and grey and orange for the other, or something like that).

We appreciate the reviewer's concern, and in fact we initially went for different colors for each context (red red or blue). In the end we opted to use a single colour (orange) because it would dichotomize the stimuli as relevant vs irrelevant which is the main message of our paper. In other words, red and blue would be coding for the same thing (relevant stimulus), so we opted to colour them the same way.

REVIEWERS' COMMENTS

Reviewer #1 (Remarks to the Author):

The authors have addressed most of my concerns. A few minor points:

(1) In my initial review, I had suggested that the authors state the caveat (for the decoding analysis) that the neural populations are not simultaneously recorded. This was mostly just for readers to understand that the y-axis values on the decoding performance cannot be trusted. The authors now write that "[results] are qualitatively similar when decoding from simultaneously recorded populations", and perform an analysis that is done on subensembles, many of which are single neurons. That's a bit of an odd twist. Maybe try to formulate this a bit better, and also please explain the error bars in Figure S6b.

(2) I still think it would help readers understand the modeling results better if certain assumptions were spelled out better. For instance, it is not necessary for PFC to both act as receiver of the relevant go information and providing the context (even though it matches the cited literature). Alternative configurations are possible, as long the particular subspace configurations are maintained. For me, pointing this out could help comprehensibility - what is really necessary, what are assumptions that could be modified - and is less about discussing possible other model configurations.

Reviewer #2 (Remarks to the Author):

First, I wanted to apologise to the authors for my delayed response —busy times, personally and professionally! Second, I want to thank them for their thorough responses. I only have a couple small follow up points regarding one of my previous comments (and the authors' responses)

- Comment 2(ii): The authors point out that they are computing the angles between the optimal decoding axes, which I also think is the best approach. However, it may be the case that the second best decoding axes has a different geometric relationship across conditions (parallel? Perpendicular?). To rule this out, I think the authors should show: 1) how many decoding axes there are, and how well they perform; 2) if they perform well (i.e., above chance), compute the pairwise angles between conditions across all axes pairs. Adding this in the main figure would be confusing, but I think it'd be important to

show this in the Supplement, as it seems to often be the case that there are several good decoding axes in neural state space

- Comment 4: This is a very nice analysis/result.

I also wanted to point out that I re-read the manuscript and now it's much clearer. I'd only want to suggest the authors to give an intuition for how "Output gating" is defined in the text, since it's central to the results.

Reviewer #1 (Remarks to the Author):

The authors have addressed most of my concerns. A few minor points:

(1) In my initial review, I had suggested that the authors state the caveat (for the decoding analysis) that the neural populations are not simultaneously recorded. This was mostly just for readers to understand that the y-axis values on the decoding performance cannot be trusted. The authors now write that "[results] are qualitatively similar when decoding from simultaneously recorded populations", and perform an analysis that is done on subensembles, many of which are single neurons. That's a bit of an odd twist. Maybe try to formulate this a bit better, and also please explain the error bars in Figure S6b.

We have now clarify that the "populations" can be of 1 to 13 neurons.

Crucially, our results do not depend on these methodological choices and are qualitatively similar when decoding from simultaneously recorded populations of 1 to 13 neurons (Fig. S6).

We also mention in the first paragraph that all analyses were performed on pseudo-trials:

In this study we focused on pseudo-trials (Methods) but results were qualitatively similar when analyzing simultaneously recorded neurons (Fig. S6).

(2) I still think it would help readers understand the modeling results better if certain assumptions were spelled out better. For instance, it is not necessary for PFC to both act as receiver of the relevant go information and providing the context (even though it matches the cited literature). Alternative configurations are possible, as long the particular subspace configurations are maintained. For me, pointing this out could help comprehensibility - what is really necessary, what are assumptions that could be modified - and is less about discussing possible other model configurations.

When discussing the network configuration, we now added the sentence ***"For instance, it is not necessary for PFC to both act as receiver of the relevant go information and provide the context"***

Reviewer #2 (Remarks to the Author):

First, I wanted to apologise to the authors for my delayed response —busy times, personally and professionally! Second, I want to thank them for their thorough responses. I only have a couple small follow up points regarding one of my previous comments (and the authors' responses)

- Comment 2(ii): The authors point out that they are computing the angles between the optimal decoding axes, which I also think is the best approach. However, it may be the case that the second best decoding axes has a different geometric relationship across conditions (parallel? Perpendicular?). To rule this out, I think the authors should show: 1) how many decoding axes there are, and how well they perform; 2) if they perform well (i.e., above chance), compute the pairwise angles between conditions across all axes pairs. Adding this in the main figure would be confusing, but I think it'd be important to show this in the Supplement, as it seems to often be the case that there are several good decoding axes in neural state space

When trying to decode from neural activity orthogonal to the optimal decoder we found that the second optimal decoder of relevant and irrelevant stimulus was slightly above chance. These the 2nd optimal decoding axis of the relevant and irrelevant stimuli were more orthogonal to each other than the optimal decoder. This analysis was added to Figure S6.

- Comment 4: This is a very nice analysis/result.

Thank you!

I also wanted to point out that I re-read the manuscript and now it's much clearer. I'd only want to suggest the authors to give an intuition for how "Output gating" is defined in the text, since it's central to the results.

We have a section of methods detailing the analysis of output gating and we are now referencing it on every mention of "output-gating".